# PP-GWAS: Privacy Preserving Multi-Site Genome-wide Association Studies

Arjhun Swaminathan [1,2] ✉, Anika Hannemann[3,4,6], Ali Burak Ünal[1,2,7], Nico Pfeifer[2,5] & Mete Akgün[1,2] ✉

Genome-wide association studies help uncover genetic influences on complex traits and diseases. Importantly, multi-site data collaborations enhance the statistical power of these studies but pose challenges due to the sensitivity of genomic data. Existing privacy-preserving approaches to performing multi-site genome-wide association studies rely on computationally expensive cryptographic techniques, which limit applicability. To address this, we present PP-GWAS, a privacy-preserving algorithm that improves efficiency and scalability while maintaining data privacy. Our method leverages randomized encoding within a distributed framework to perform stacked ridge regression on a linear mixed model, enabling robust analysis of quantitative phenotypes. We show experimentally using real-world and synthetic data that our approach achieves twice the computational speed of comparable methods while reducing resource consumption.

Genome-wide association studies (GWAS) have emerged as a critical instrument for discerning the genetic components that underlie complex biological traits and diseases. By investigating differences in allele frequencies of genetic variants, particularly single-nucleotide polymorphisms (SNPs), between ancestrally similar individuals exhibiting distinct phenotypic traits, GWAS have highlighted numerous genomic risk loci associated with a variety of diseases and characteristics[1–3]. The power of these studies is realized especially when multiple datasets are collaboratively analyzed, as such joint efforts have consistently revealed a broader spectrum of associations than when individual datasets are studied in isolation[4,5].

Nevertheless, despite the potential advantages, multi-site dataset collaborations in the realm of GWAS are rarely pursued. This can be attributed predominantly to stringent institutional policies and regulations, such as the General Data Protection Regulation (GDPR) in the European Union, which act as obstacles to the sharing of sensitive genetic data[6]. The emphasis on privacy is not exclusive to the European Union. Other jurisdictions, including in Africa, have started to bolster privacy protections as a response to the growing awareness of

the potential misuse of sensitive data[7]. This global move towards stringent data protection presents an important discussion: on one hand, there is the undeniable potential of collaborative GWAS in advancing medical science, and on the other, there is the indispensable need to safeguard individual privacy[8].

A well-established technique for multi-site data collaborations in the context of genomic studies is meta-analysis[9], which combines summary statistics from independent GWAS to identify associations in the total combined sample. Although it can mitigate some privacy concerns by avoiding the direct exchange of individual-level data, meta-analysis is susceptible to biases arising from heterogeneous cohorts, varying sample sizes, and differing imputation or phenotyping strategies[10,11]. These discrepancies can impair the consistency of estimated genetic effects, highlighting the need for approaches that analyze data jointly while still preserving privacy.

Thus, a growing interest[12–15] in secure computation for collaborative multi-site GWAS has led to solutions such as[12], S-GWAS[16], FAMHE[17], and SF-GWAS[18]. S-GWAS[16] was one of the first practically feasible frameworks designed for large-scale data. It relies on a secure

[1]Medical Data Privacy and Privacy-preserving Machine Learning (MDPPML), University of Tübingen, Tübingen, Germany. [2]Institute for Bioinformatics and Medical Informatics (IBMI), University of Tübingen, Tübingen, Germany. [3]Dept. of Computer Science, Leipzig University, Leipzig, Germany. [4]Center for Scalable Data Analytics and Artificial Intelligence (ScaDS.AI) Dresden/Leipzig, Leipzig, Germany. [5]Methods in Medical Informatics, University of Tübingen, Tübingen, Germany. [6]Present address: Zurich University of Applied Sciences, School of Engineering, Zurich, Switzerland. [7]Present address: Intelligent Vehicles Lab, Delft University of Technology, Delft, Netherlands. ✉e-mail: arjhun.swaminathan@uni-tuebingen.de; mete.akguen@uni-tuebingen.de

multiparty computation (MPC)[19] backbone: multiple computational nodes hold secret shares of the original data and cooperate in such a way that no individual's genetic or phenotypic information is exposed. A key factor in S-GWAS's efficiency is its adaptation of Beaver triples, a widely used multiplication technique in MPC, generalized to handle exponentiation and other higher-order operations essential to genomic analyses. Further, S-GWAS uses pseudo-random generators to help mitigate the typical communication overhead associated with MPC. To address population stratification, S-GWAS employs random projection methods that reduce the dimension of genotype matrices before running principal component analysis (PCA). Operating under a non-colluding semi-honest model, S-GWAS is best suited for quantitative traits; for binary traits, the authors propose a two-stage procedure: first, Cochran–Armitage trend tests narrow down candidate variants, then logistic regression is applied only to that reduced subset.

FAMHE[17] subsequently explored the use of homomorphic encryption (HE)[20] to achieve privacy-preserving GWAS. In FAMHE, each computational node can run operations locally on its unencrypted data before encrypting intermediate results with HE and sharing them. These intermediate encrypted values are then aggregated and redistributed for further computation. While FAMHE eliminates many of MPC's communication hurdles and excels at additive and multiplicative operations, it contends with considerable computational overhead and must approximate non-linear operations (such as those required for logistic regression), diminishing precision.

Building on both S-GWAS and FAMHE, SF-GWAS[18] strengthens the architecture further by integrating federated learning principles alongside MPC and MHE. It addresses one of the main drawbacks of purely homomorphic strategies-namely, the difficulty of non-linear operations like division and comparisons−by partitioning the analysis pipeline. Homomorphic encryption handles additions and multiplications on encrypted data, while dedicated MPC routines perform divisions, comparisons, and other operations more cumbersome for MHE alone. SF-GWAS also provides two key workflows: a PCA-based approach that uses linear regression for quantitative traits and logistic regression for binary traits, and a linear mixed model (LMM)-based workflow inspired by REGENIE[21], which relies exclusively on linear regression for quantitative traits. As a result, SF-GWAS offers improvements in terms of practical performance and versatility compared to earlier methods, while still preserving data privacy under an all-but-one semi-honest adversarial model.

However, despite these advances in privacy-preserving multi-site GWAS, methods relying on MPC and MHE still pose practical challenges, which become especially pronounced when handling large-scale datasets. MPC often requires frequent communication among participants and may need reconfiguration when new data providers join, whereas MHE demands specialized on-premise computational resources that many healthcare institutions may lack[22,23].

With the challenges presented, our work seeks to integrate GWAS into a distributed architecture where a single third-party helper node is tasked with helping data providers carry out multi-site GWAS in a privacy-preserving manner. Hence, we introduce PP-GWAS as an alternative to state-of-the-art solutions that aim to perform association tasks for quantitative traits with high accuracy and reduced computational strain. We evaluate our method against S-GWAS[16] and its more powerful successor, SF-GWAS[18]. Unlike S-GWAS and SF-GWAS, which utilize MPC and MHE to perform secure multi-site GWAS, our method relies on randomized encoding in a distributed architecture, resulting in improved efficiency and lower computational demands.

Randomized encoding[24,25] achieves privacy preservation by obfuscating data in a lower/higher dimensional space. The encoding is dependent upon the analysis performed on the data, and hence establishing security depends on the encoding used[26]. This translates into a dynamic challenge of identifying potential vulnerabilities and attacks rather than proving robustness from the outset. In our work,

we use randomized encoding to obfuscate the data, as in other applications such as[27–30]. By employing this approach, we shift the computational burden away from the intensive multi-round communication or specialized hardware requirements typical of MPC and MHE, making our approach more accessible to resource-limited healthcare institutions.

We adapt a well-established centralized GWAS algorithm based on Linear Mixed Models, REGENIE[21], into a distributed and privacy-preserving setting. We do this since REGENIE is particularly adept at managing large-scale datasets. It employs a two-step methodology, where one first performs ridge regression on the whole genome data to arrive at a smaller space of predictions. Subsequently, another round of ridge regression is performed on these predictions in a stacked fashion, and the SNPs are individually tested.

In this work, we evaluate PP-GWAS against its alternatives on both synthetic data, generated using pysnptools[31], and two real-world datasets: a Bladder Cancer Risk dataset and an Age-Related Macular Degeneration (AMD) dataset. Our empirical findings highlight a notable advancement in both scalability and execution speed, with PP-GWAS performing nearly twice as fast as SF-GWAS. Most importantly, these speeds are achieved utilizing computational resources that are considerably less than what SF-GWAS necessitates, making our approach more pertinent to real-world scenarios. Moreover, the accuracy of our GWAS results is validated against REGENIE and meta-analysis, ensuring comprehensive evaluation. Further, in terms of the adversarial conditions, SF-GWAS operates within an all-but-one semi-honest adversarial model, and incorporates an external node designated as a helper. However, the potential for malicious intent from this external node remains ambiguous. In contrast, our approach distinctly outlines the role of the external node, categorizing it as both non-colluding and semi-honest. This clear description not only ensures the method's precision, but also aligns with standard privacy-enhancing techniques in distributed frameworks[8,32].

## Results
### Experimental setup
Most of our experiments, unless mentioned otherwise, were conducted on a state-of-the-art high-performance computing (HPC) cluster. Each node within this HPC environment was equipped with an Intel XEON CPU E5-2650 v4, complemented by 256 GB of memory and a 2 TB SSD storage capacity. We employed Python as the primary programming language, taking advantage of Intel's Math Kernel Library (MKL) for high-demand computational tasks. A dynamic core allocation strategy was utilized for MKL-based operations, enhancing computational efficiency and throughput. To ensure the robustness and reproducibility of our experimental findings, each experiment was conducted five times, and the results were averaged. Error bars in the runtime figures represent deviations from these multiple iterations, reflecting the consistency of our measurements. Our runtime comparisons prominently include SF-GWAS (PCA-based), with the reported execution times for SF-GWAS sourced directly from their original publication[18].

The architecture of our experimental system was distributed across multiple nodes of the HPC cluster. The server was allowed to access 128 GB of memory for all experiments, while the other individual nodes utilized memory variably, utilizing up to a maximum of 32 GB unless specified otherwise. Such a configuration is reminiscent of real-world scenarios where computational tasks are commonly outsourced by medical and research institutions (Fig. 1). This design also mirrors the setup described in SF-GWAS, though with more constrained memory allocations for the nodes.

Communication between the server and the nodes was facilitated through socket programming, implemented using TCP connections. Each node established a connection to the server through a unique port. The server, leveraging multiprocessing capabilities, managed

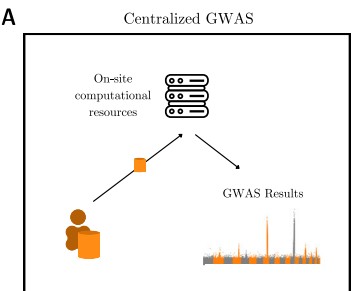
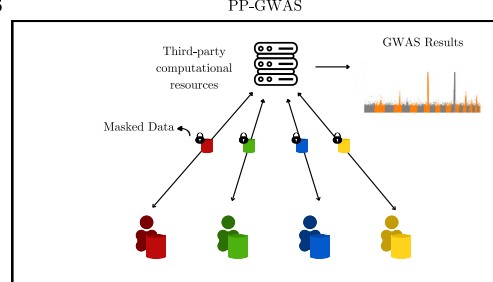

**Fig. 1 | Comparison of centralized GWAS and PP-GWAS. A** A centralized approach is depicted where a single institution, such as a hospital or research institute, utilizes on-site computational resources to conduct GWAS on its local data. **B** A distributed model is illustrated, in which multiple entities collaborate to perform GWAS on a combined dataset. This is achieved without sharing the local data and by leveraging a third-node service to facilitate computations.

simultaneous data exchanges with multiple nodes. This approach ensured real-time interactions and minimized node idleness. The communication was characterized by a round-trip latency of 0.249 ms, with the TCP window size set at the default 128 kByte. For matrix operations, especially involving large sparse datasets, we integrated the sparse-dot-mkl library[33].

In summary, our experimental design was crafted to leverage the capabilities of the HPC cluster, drawing parallels with the setup detailed in SF-GWAS. By optimizing computational resources and ensuring efficient communication protocols, our aim was to create a versatile system, adept at addressing the stringent demands of privacy-preserving genome-wide association studies.

## Synthetic data generation

Synthetic data for our experiments was generated using the pysnptools library[31] and was simulated to resemble quantitative traits. The population structure was set at 0.1, and the degree of family relatedness was fixed at 0.25. This synthetic data was horizontally partitioned across the nodes. The synthetic datasets varied widely in size, with sample sizes ranging from 9178 to 275,000, SNP counts ranging from 580,000 to 2451, 176, and the number of covariates ranging from 2 to 40.

## Real datasets

Two real genomic datasets were used for the experiments: A Bladder Cancer Risk dataset (13,060 Samples, 467,172 SNPs) (dbGaP Study Accession: phs000346.v2.p2)[34-36], and an Age-Related Macular Degeneration dataset (22,683 Samples, 508,740 SNPs) (dbGaP Study Accession: phs001039.v1.p1)[37]. Access to these datasets was secured through the dbGaP platform, adhering to the necessary procedural requirements. These datasets were further imputed for missing data using Beagle[38]. Since our GWAS algorithm is tailored for quantitative data, we treat these real datasets as if they were quantitative. Both dbGaP releases include standard subject-level covariates. For the Bladder Cancer Risk dataset, age, sex, and study-center indicators (capturing platform information) were available; for the AMD dataset, age and sex were available across cohorts. We included these as covariates in our analyses.

Both the synthetic and real datasets were stored blockwise in the .npz format, which our code is designed to read.

## Quality control

For both the synthetic and real datasets, a series of preprocessing steps was performed to ensure appropriate data quality. These are a part of the algorithm, and are included in the runtime analysis in the subsequent section. Genotypes with a missing rate exceeding 0.1 were filtered out. Further, only alleles with a minor allele frequency greater than 0.05 were retained. Lastly, a Hardy–Weinberg equilibrium chi-squared test statistic threshold of 23.928 (corresponding to a *p*-value of $10^{-6}$) was applied. These preprocessing measures were securely executed on the whole data by the nodes using standard addition-based randomized encoding techniques.

## Accuracy analysis

To rigorously evaluate the accuracy of PP-GWAS, it was essential to conduct a comparative analysis against the well-established unencrypted plaintext GWAS algorithm, REGENIE. Using Pearson's *r*-square coefficient as a measure of accuracy between the negative log of the *p*-values, PP-GWAS demonstrated robust performance on real-world datasets. Specifically, the Pearson correlation of $-\log_{10}(p)$ between our method and REGENIE across both datasets was $r^2 = 0.999999$–$1.00$ (df $= M-2$), $P \sim 0$, 95% CI [0.999999, 1.00], where $M$ is the number of SNPs. These outcomes, illustrating high correlation with the plaintext benchmarks, are detailed in Fig. 2. This comparison highlights the capability of PP-GWAS to maintain genetic association analysis accuracy while ensuring data privacy.

## Scalability analysis of PP-GWAS with simulated data

The ability to maintain both computational efficiency and accuracy with large-scale data is a critical challenge in genome-wide association studies. This section provides a comparative analysis between PP-GWAS and SF-GWAS, focusing on performance under various conditions.

To facilitate a direct comparison, we utilized a simulated dataset designed similarly to those in SF-GWAS's scalability analysis. We consider four primary factors in our scalability analysis: the number of computational nodes, the SNP count within the genomic data, the number of covariates within the genomic data, and the sample sizes managed by each node. Incremental increases in each of these factors allow us to observe and quantify the performance implications on PP-GWAS.

Our initial evaluation focuses on the algorithm's performance in response to an increasing number of nodes. With a test dataset comprising 9178 samples and 612,794 SNPs, we assess the algorithm's distributed computation capabilities. Performance outcomes, as shown in Fig. 3A, indicate that PP-GWAS maintains a linear performance with an increasing number of nodes.

We then explore the scalability in relation to SNP counts, with a fixed configuration of two nodes and 9178 samples. Addressing the large-scale nature of many genomic datasets, PP-GWAS's performance remains superior to that of SF-GWAS, as depicted in Fig. 3B.

Next, we explore the scalability in relation to the number of covariates, with a fixed genomic dataset size of 9178 samples and 612,794 SNPs. We note in Fig. 3C that the runtime is unaffected by an increase in the number of covariates since projecting out covariates is done early in our methodology, and is a cheaper operation as opposed to working with the whole genomic dataset.

A

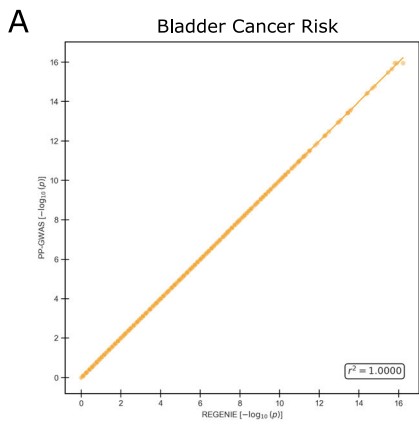

B

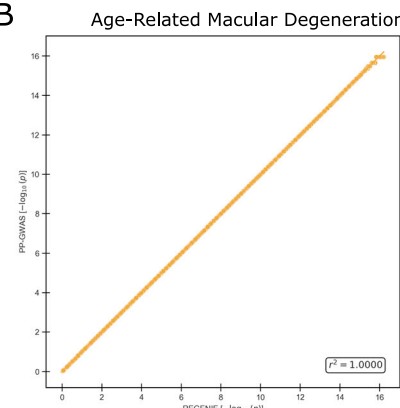

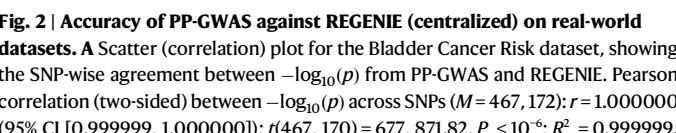

**Fig. 2 | Accuracy of PP-GWAS against REGENIE (centralized) on real-world datasets. A** Scatter (correlation) plot for the Bladder Cancer Risk dataset, showing the SNP-wise agreement between $-\log_{10}(p)$ from PP-GWAS and REGENIE. Pearson correlation (two-sided) between $-\log_{10}(p)$ across SNPs ($M = 467,172$): $r = 1.000000$ (95% CI [0.999999, 1.000000]); $t(467,170) = 677,871.82$, $P < 10^{-6}$; $R^2 = 0.999999$.

**B** Scatter (correlation) plot for the Age-Related Macular Degeneration (AMD) dataset, depicting the analogous correlation. Pearson correlation (two-sided) between $-\log_{10}(p)$ across SNPs ($M = 508,740$): $r = 1.000000$ (95% CI [0.999999, 1.000000]); $t(508,738) = 577,736.14$, $P < 10^{-6}$; $R^2 = 0.999998$. No multiple-testing correction was applied to these correlation tests.

Lastly, we examine how sample size affects PP-GWAS's scalability. Keeping the number of nodes at two and SNPs constant at 612,794, we increment the sample size and analyze the impact. The performance of PP-GWAS against increasing sample sizes is demonstrated in Fig. 3D.

In conclusion, the scalability analysis underscores PP-GWAS's capability to efficiently manage increased computational demands across various dimensions. This is instrumental for its application in extensive genetic association studies.

## Adaptability to large-scale data
To address the challenge of scaling PP-GWAS for large-scale genomic analyses, we conducted experiments using synthetic datasets, given the inaccessibility of datasets such as the UK Biobank and eMERGE. For simulations other than the UK Biobank scale, the system was configured with the central server being allocated 256 GB of RAM and six participant nodes, each provided with 56 GB of RAM. In contrast, for the UK Biobank-sized experiments—which comprised 275,000 samples and 580,000 SNPs—we leveraged deNBI Cloud resources, which consisted of vastly different hardware as compared with what is offered by Google Cloud. Due to the technical constraints, our configuration employed a modified setup with four client nodes, each assigned 256 GB of RAM, alongside a central server equipped with 700 GB of RAM.

Under the deNBI Cloud setup simulating the UK Biobank configuration, PP-GWAS completed the analysis in 2 days 18 h and 49 min, while the simulation configured to represent the eMERGE dataset finished in 8 h and 7 min, as illustrated in Fig. 3E. These results provide a clear assessment of PP-GWAS's scalability across large-scale dataset sizes and different computational environments. Moreover, under linear interpolation, we expect PP-GWAS to complete the UK Biobank dataset sized experiments in 3 days 5 h and 30 min if we had the same computational resources and six-node configuration as SF-GWAS.

## Memory efficiency and communication cost analysis
In the realm of privacy-preserving GWAS, PP-GWAS algorithm presents a notable shift from SF-GWAS, especially in terms of memory efficiency and communication costs. This section goes into how these two critical factors play out in the implementation and scalability of PP-GWAS.

*Memory efficiency*: A key strength of PP-GWAS lies in its significantly reduced RAM requirements compared to SF-GWAS, as discussed in Fig. 4B. This aspect is particularly advantageous for settings with limited computational resources, such as smaller research

institutions or medical facilities. By lowering the memory demands, PP-GWAS enables these organizations to partake in large-scale genetic studies without the need for extensive hardware upgrades. This improvement in memory efficiency is instrumental in democratizing GWAS, allowing for wider and more inclusive research participation.

*Communication costs*: As seen in Fig. 4A, while PP-GWAS requires higher communication overhead than SF-GWAS when the number of computational nodes is low, this increase is a strategic trade-off. Specifically, the communication demands in PP-GWAS rise linearly and predictably, in contrast to the exponential growth experienced by SF-GWAS as the number of nodes increases. This makes PP-GWAS a more accessible option for many institutions, especially in an era where digital connectivity often surpasses the availability of advanced computational resources. Furthermore, the distributed nature of the PP-GWAS algorithm reduces the number of communication rounds, alleviating some of the burdens seen in SF-GWAS.

## Performance in LAN and WAN settings
Evaluating the performance of PP-GWAS across different network configurations is essential to its applicability in real-world scenarios. Using simulated data, we compared the performance of PP-GWAS to SF-GWAS in both local-area network (LAN) and wide-area network (WAN) settings using Google Cloud.

For these experiments, we replicated the network setup from SF-GWAS. In the WAN configuration, three computational nodes were distributed across geographically distant regions: two clients located in Iowa (us-central1) and London (europe-west2), and the server in North Virginia (us-east4). For the LAN configuration, all nodes were placed in Northern Virginia (us-east4). We progressively scaled the dataset size, using sample sizes ranging from 9178 to 36,712, with 612,794 SNPs. The round-trip latency matched the SF-GWAS setup, measuring 0.3 ms in the LAN and up to 100 ms in the WAN.

In addition to runtime, we measured the total volume of data transferred between a client and the server in each experiment to understand the communication efficiency of PP-GWAS. The total data transferred increased with sample size: 9178 samples (188.9 GB), 18,356 samples (377.6 GB), 27,534 samples (566.5 GB), and 36,712 samples (755.6 GB). These values provide an estimate of the communication overhead in general. Figure 4(C) illustrates the runtime performance of PP-GWAS in both LAN and WAN settings relative to SF-GWAS, highlighting its adaptability to varying network conditions.

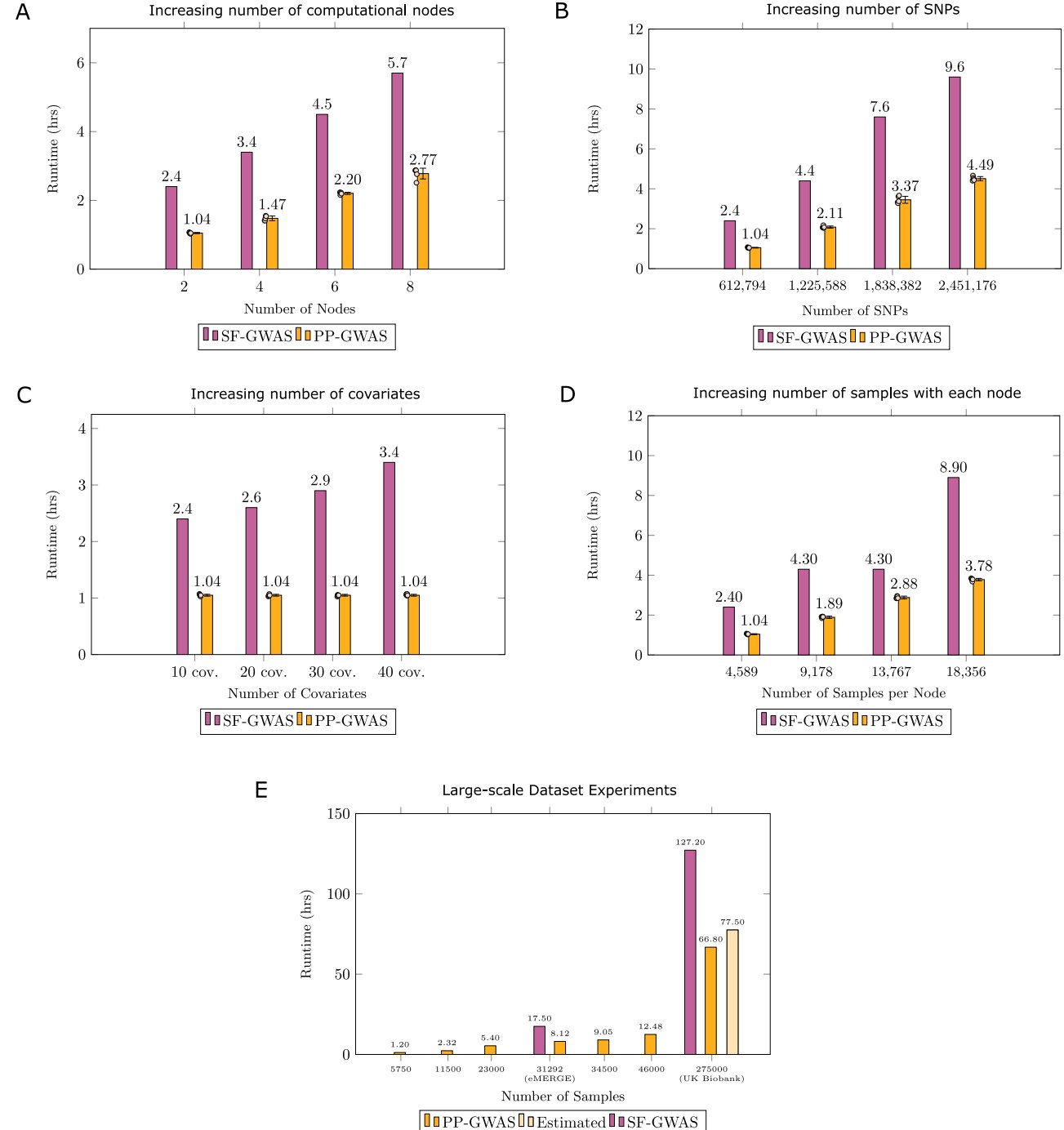

**Fig. 3 | Scalability analysis of PP-GWAS. A** Comparison of total computational times for SF-GWAS and PP-GWAS, analyzing a dataset (9178 samples × 612,794 SNPs) across a varying number of participating institutions. **B** Comparison of total computational times for SF-GWAS and PP-GWAS, analyzing a dataset with 9178 samples across two participating institutions, and an increasing number of SNPs. **C** Comparison of total computational times for SF-GWAS and PP-GWAS, analyzing a dataset with 9178 samples and 612,794 SNPs across two participating institutions, and an increasing number of covariates. **D** Comparison of total computational times for SF-GWAS and PP-GWAS, analyzing a dataset with 612,794 SNPs across two participating institutions, and an increasing number of samples. **E** Comparison of total computational times for PP-GWAS and SF-GWAS when applied to large-scale datasets equivalent in size to the eMERGE and the UK Biobank datasets. Source data are provided as a Source Data file. *Data presentation and statistics (3A–D):* Bars show mean values, and error bars show ± standard deviation of five independent runs of PP-GWAS; all individual runs are overlaid as jittered dots to display the distribution. Statistical summaries are derived from technical (not biological) replicates because the objective is to quantify computational runtime variability.

## Performance Evaluation against Meta-Analysis

Here, we evaluate the performance of meta-analysis, which relies on combining individual node association results, and compare it to both centralized GWAS (REGENIE) and PP-GWAS. The comparison is conducted using two real-world datasets, both treated like quantitative data: the Bladder Cancer dataset (Fig. 5) and the AMD dataset (Fig. 6).

For meta-analysis, we utilized PLINK with configurations involving 2–6 parties, while PP-GWAS was evaluated with 6 parties. Unlike meta-

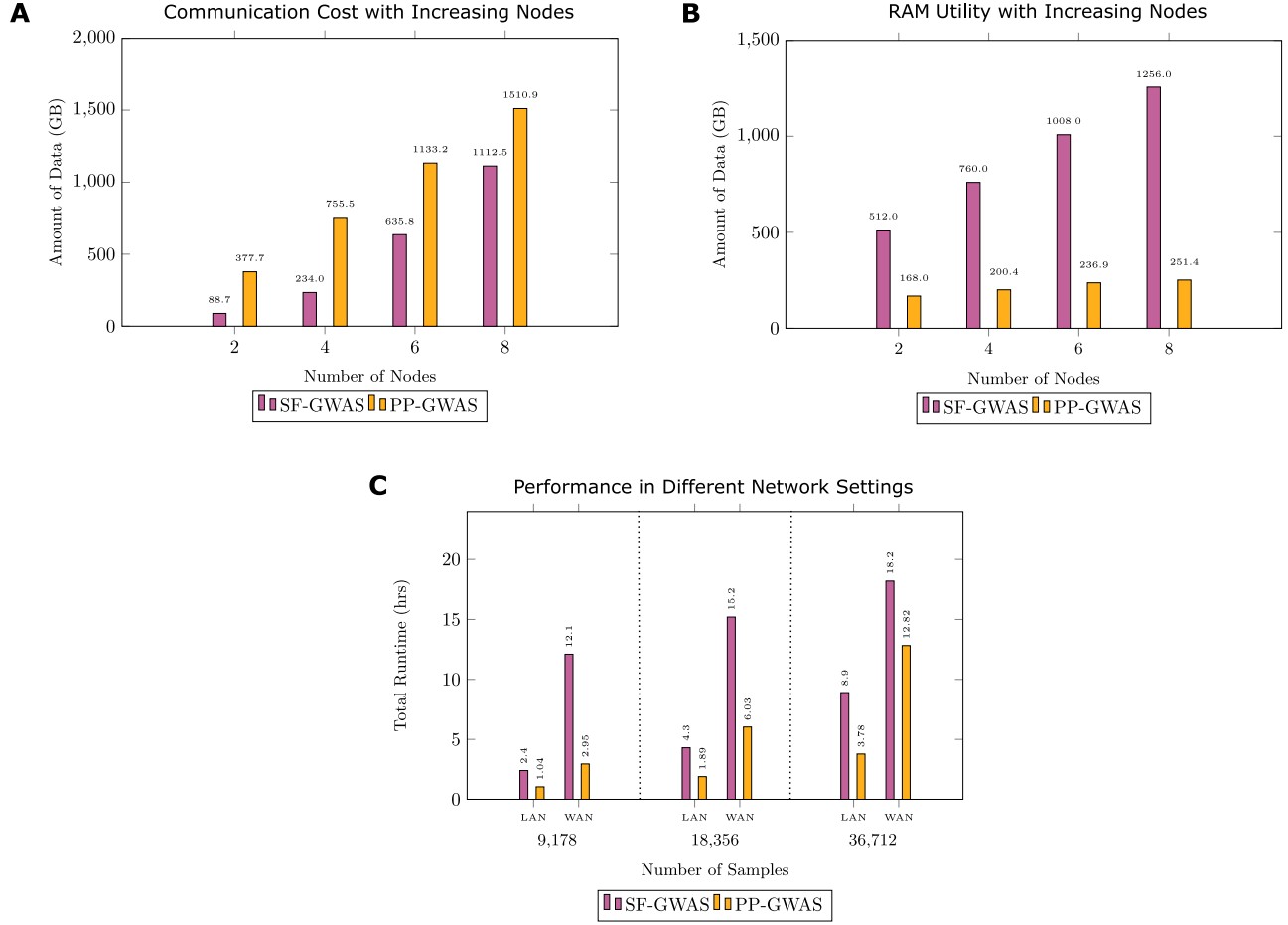

**Fig. 4 | Communication cost, memory usage, and performance across network settings. A** Comparison of communication cost (in GB) across an increasing number of computational nodes, analyzing a genetic dataset consisting of 9178 samples and 612,794 SNPs. **B** Comparison of RAM utility (in GB) across an increasing number of computational nodes, analyzing a genetic dataset consisting of 9178 samples and 612,794 SNPs. **C** Comparison of total runtimes of PP-GWAS and SF-GWAS under both LAN and Trans-Atlantic WAN settings, with varying sample sizes. Source data are provided as a Source Data file.

analysis, PP-GWAS's performance is independent of the number of parties and consistently achieves an $r^2$ accuracy of 1, demonstrating its robustness.

Our findings highlight that as data becomes more fragmented across an increasing number of parties, the performance of meta-analysis deteriorates. This decline occurs because each node works with progressively smaller sample sizes, leading to worse individual-level summary statistics. In contrast, PP-GWAS maintains high accuracy regardless of the degree of data partitioning.

To further illustrate these performance differences, we conducted additional experiments using a simulated dataset comprising 20, 000 samples and 500, 000 SNPs, distributed across 6 computational nodes. We applied REGENIE, PP-GWAS, and meta-analysis to this dataset and generated the resulting Manhattan plots. We note in Fig. 7 that REGENIE serves as the reference. PP-GWAS exhibits a near-identical distribution. Minor variations in peak cut-offs can be attributed to numerical differences introduced by floating-point arithmetic, which do not impact overall accuracy. In contrast, meta-analysis exhibits weaker association signals and increased variance across detected loci.

These results further validate the advantages of PP-GWAS, demonstrating its ability to achieve accuracy comparable to centralized GWAS while preserving data privacy. Importantly, its robustness to data partitioning highlights its suitability for collaborative genomic studies.

## Discussion

In this study, we introduced PP-GWAS, a privacy-preserving distributed framework designed to perform multi-site genome-wide association studies on quantitative data. Our extensive comparative analysis demonstrates that PP-GWAS maintains genetic association analysis accuracy equivalent to traditional centralized methods, in the analysis of real-world datasets such as the Bladder Cancer Risk dataset and the age-related macular degeneration (AMD) dataset.

PP-GWAS excels in scalability and adaptability when tested against the state-of-the-art privacy-preserving GWAS algorithm SF-GWAS[18]. Through evaluations with varying numbers of computational nodes, SNP counts, and sample sizes, our framework demonstrated a consistent linear performance increase, proving its effectiveness in multi-site GWAS. This scalability is essential for accommodating the expanding size and diversity of genomic datasets in real-world scenarios, making PP-GWAS a stable solution even under the constraints of limited computational resources. Furthermore, the adaptability of PP-GWAS was tested using synthetic datasets as proxies for large-scale real datasets, predicting feasible processing times for extensive databases such as the eMERGE and the UK Biobank datasets.

Another significant advancement is in memory efficiency and communication costs. PP-GWAS considerably reduces the RAM requirements, enabling institutions with constrained computational resources to participate in genomic research. While it necessitates higher communication overhead than SF-GWAS with fewer nodes, this

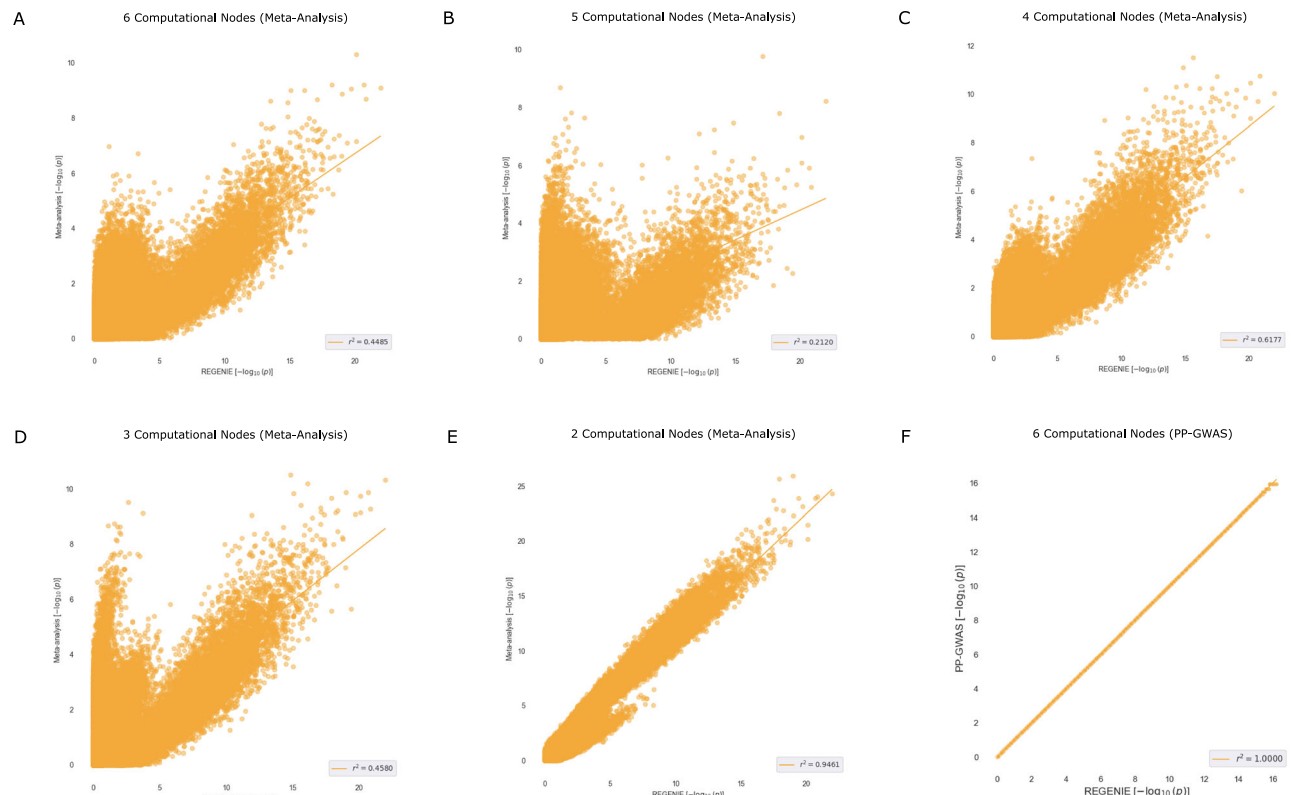

**Fig. 5 | Accuracy comparison of PP-GWAS against meta-analysis. A–F** Scatter correlation plots to compare the performance of meta-analysis with varying computational nodes against PP-GWAS on the Bladder Cancer Risk dataset. Pearson correlation (two-sided) between the $-\log_{10}(p)$ values is reported. No multiple-testing correction was applied to these correlation tests.

overhead progresses in a predictable and manageable linear fashion, which is a strategic compromise for achieving greater computational and memory efficiency. Further, since the communication overhead for SF-GWAS increases exponentially, we expect to perform better with more nodes. This trade-off ensures applicability across a broader spectrum of research environments, from hospitals to smaller research institutions.

In addition, our experiments investigating network performance further highlight the strengths of PP-GWAS. Using both local-area network (LAN) and wide-area network (WAN) settings on Google Cloud, we observed that PP-GWAS maintains competitive performance across varying network conditions. These findings confirm the potential for deployment in diverse real-world settings, from localized institutional networks to globally distributed research collaborations.

Our performance evaluation against traditional meta-analysis approaches highlights the superiority of PP-GWAS in terms of accuracy and reliability. While meta-analysis suffers from deteriorating performance as the number of collaborating parties increases, owing to progressively smaller sample sizes per node, PP-GWAS consistently retains accuracy. This performance, even under substantial data fragmentation, underscores the efficacy of PP-GWAS as a powerful solution for collaborative genomic research.

### Limitations
PP-GWAS operates on datasets that may be generated by different sites without joint genotyping. In such settings, platform- and pipeline-specific biases can induce variant-level discrepancies. We mitigate global batch effects via harmonization (shared positions, alleles, strand, and rsIDs), perform global quality control that retains rare variants present at any participating site, and remove covariate effects using covariate projection, which includes site, platform/pipeline, and

batch indicators. These steps, which are standard even in centralized analyses where data is pooled from different sources[21,39–44], are effective for single-variant association but do not eliminate all effects of technical heterogeneity.

PP-GWAS, as well as other state-of-the-art privacy-preserving distributed GWAS, would be most effective when upstream variant calls are produced within a unified framework. The privacy-preserving way to achieve this is a distributed joint-genotyping layer that accounts for platform differences during variant calling without centralizing raw data. Designing such a layer e.g., using secure aggregation, multi-party computation, homomorphic encryption, or trusted hardware remains an important direction for future research.

Finally, we do not advocate centralizing or sharing raw genotypes for joint genotyping and then returning to a privacy-preserving distributed GWAS workflow. Were genotypes to be shared, the core rationale for privacy-preserving analyses would be undermined. PP-GWAS is therefore intended either (i) for non-jointly genotyped settings with the above mitigations and explicit technical covariates, acknowledging residual confounding may persist, or (ii) to be composed with a privacy-preserving distributed joint-genotyping layer.

## Methods
This research complies with all relevant ethical regulations. Access to dbGaP datasets used in this study, phs000346.v2.p2 and phs001039.v1.p1, was authorized by the NIH dbGaP Data Access Committees (NCI DAC for phs000346; NEI DAC for phs001039).

### Linear mixed models in genome-wide association studies
With regards to GWAS, the application of linear mixed models (LMMs) has emerged as a fundamental approach for deciphering the intricate

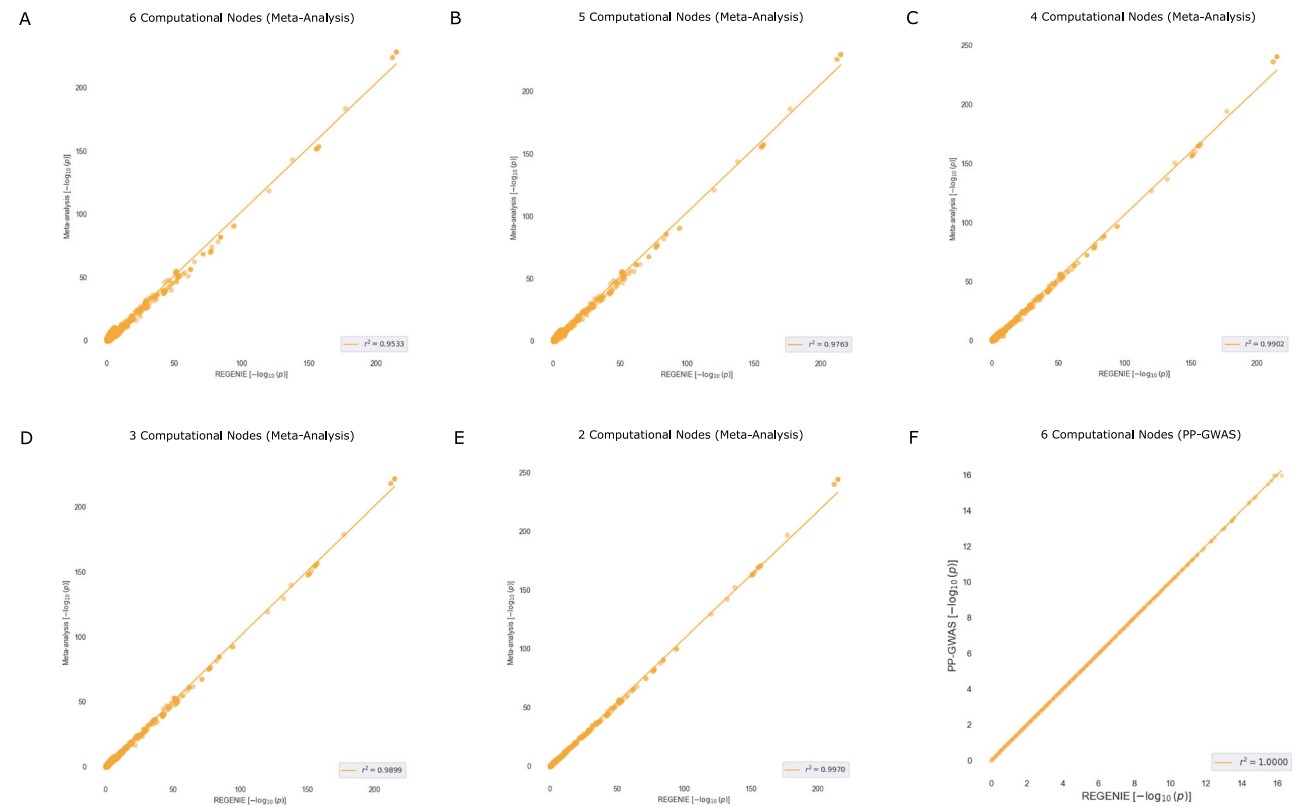

**Fig. 6 | Accuracy comparison of PP-GWAS against meta-analysis. A–F** Scatter correlation plots to compare the performance of meta-analysis with varying computational nodes against PP-GWAS on the AMD dataset. Pearson correlation (two-sided) between the $-\log_{10}(p)$ values is reported. No multiple-testing correction was applied to these correlation tests.

genetic underpinnings of various phenotypes. A standard linear mixed model used for GWAS is described below:

$$\mathbf{y} = \beta_{\text{test}} \mathbf{x}_{\text{test}} + \mathbf{Z}\boldsymbol{\alpha} + \mathbf{g} + \mathbf{e}. \tag{1}$$

Here $\mathbf{y}$ represents the phenotype vector of $N$ individuals while $\mathbf{x}_{\text{test}}$ encapsulates the minor allele dosages of the variant being tested, represented as 0, 1, or 2, signifying reference-homozygous, heterozygous, and alternate homozygous alleles, respectively. This is represented as a column vector, similar to $\mathbf{y}$, which are both standardized initially to have mean zero and unit standard deviation. An $N \times C$ matrix $\mathbf{Z}$ accounts for other confounding factors. The polygenic effect $\mathbf{g}$ includes multiple small-effect size variants. Specifically, $\mathbf{g} = \mathbf{X}\boldsymbol{\beta}$, with $\mathbf{X}$ representing the standardized genotypes of $m$ variants. Environmental effects denoted by $\mathbf{e}$, is modeled as Gaussian noise.

Both $\mathbf{x}_{\text{test}}$ and $\mathbf{y}$ are standardized to have zero mean and unit variance. The model incorporates fixed effects ($\beta_{\text{test}}$ and $\boldsymbol{\alpha}$) and random effects ($\mathbf{g}$ and $\mathbf{e}$). The genetic effect uses what is called the kinship matrix $\mathbf{K} = \frac{1}{m}\mathbf{X}\mathbf{X}^{\top}$, with $\boldsymbol{\beta} \sim \mathcal{N}(\mathbf{0}, (\sigma_g^2/m)\mathbf{I}_{m \times m})$, leading to $\mathbf{g} \sim \mathcal{N}(\mathbf{0}, \sigma_g^2\mathbf{K})$. The environmental effect is modeled as $\mathbf{e} \sim \mathcal{N}(\mathbf{0}, \sigma_e^2\mathbf{I}_{n \times n})$. The variance components $\sigma_g^2$ and $\sigma_e^2$ represent the polygenic and environmental variances, respectively.

The model's validity is assessed by testing the null hypothesis $H_0: \beta_{\text{test}} = 0$ for each variant, thus identifying significant associations with the phenotype under study. A pivotal aspect of LMM implementation is the projection of covariates from phenotypes and genotypes, a technique used to remove any confounding effects. This is done by projecting the genomic matrix and the phenotype data to the null space of $Z$. The projection matrix is formalized as

$$\mathbf{P} = \mathbf{I}_n - \mathbf{Z}\left(\mathbf{Z}^{\top}\mathbf{Z}\right)^{-1}\mathbf{Z}^{\top} \tag{2}$$

Post-projection, the model assumes the form:

$$\tilde{\mathbf{y}} = \beta_{\text{test}}\tilde{\mathbf{x}}_{\text{test}} + \tilde{\mathbf{X}}\boldsymbol{\beta} + \mathbf{e} \tag{3}$$

where $\tilde{\mathbf{y}} = \mathbf{P}\mathbf{y}$, $\tilde{\mathbf{x}}_{test} = \mathbf{P}\mathbf{x}_{test}$ and $\tilde{\mathbf{X}} = \mathbf{P}\mathbf{X}$. This approach effectively removes the influence of covariates, yielding residuals that more accurately reflect the relevant genetic associations. The LMM-based $\chi^2$ test statistic, central to hypothesis testing, is given by

$$\chi^2 = \frac{\left(\tilde{\mathbf{x}}_{\text{test}}^{\top}\mathbf{V}^{-1}\tilde{\mathbf{y}}\right)^2}{\tilde{\mathbf{x}}_{\text{test}}^{\top}\mathbf{V}^{-1}\tilde{\mathbf{x}}_{\text{test}}} \tag{4}$$

where $\mathbf{V} = \hat{\sigma}_g^2\mathbf{K} + \hat{\sigma}_e^2\mathbf{I}_{n \times n}$ given the maximum likelihood estimates $\hat{\sigma}_g$ and $\hat{\sigma}_e$ of the variance parameters $\sigma_g$ and $\sigma_e$.

**Stacked ridge regression for LMM-based GWAS**

The computation of association statistics within the framework of LMMs presents a significant computational challenge. This arises primarily due to the necessity of maximum likelihood estimation of the variance parameter $\sigma_g$, which involves large matrix operations. This complexity escalates dramatically for large-scale datasets, often making the computations prohibitively resource-intensive. Traditional efforts in algorithmic development have primarily focused on

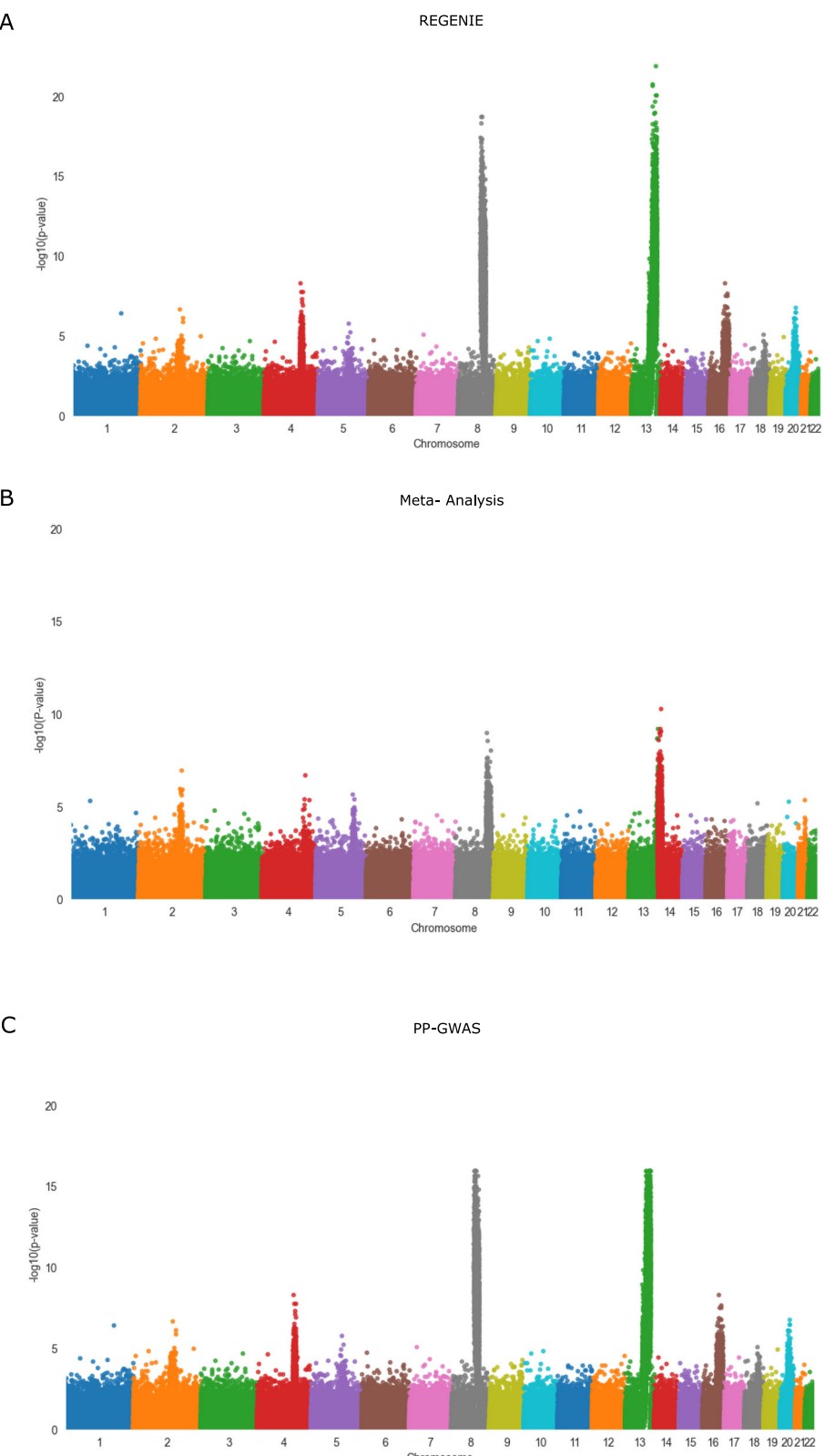

**Fig. 7 | Manhattan Plots of REGENIE, meta-analysis, and PP-GWAS.** Manhattan plots display $-\log_{10}(p)$ for single-SNP additive association tests in simulated data ($N$ = 20,000 samples, $M$ = 500,000 SNPs). **A** REGENIE. $p$-values arise from the single-SNP association testing as implemented in REGENIE; the null hypothesis is $\boldsymbol{\beta}$ = 0, the test statistic follows $\chi^2$ with df = 1, and two-sided $p$-values are reported. **B** Meta-analysis. Per-site SNP effects and standard errors are combined by fixed-effect inverse-variance meta-analysis to a pooled $Z$ statistic (with $Z^2 \sim \chi^2$ under the null hypothesis H$_0$: $\boldsymbol{\beta}$ = 0); two-sided $p$-values are reported. **C** PP-GWAS. $p$-values are computed from the distributed single-SNP association test (Box 3); the reported statistic is $\chi^2$ with df = 1 under the null hypothesis H$_0$: $\boldsymbol{\beta}$ = 0, yielding two-sided $p$-values. For all panels, $p$-values are exact and unadjusted across SNPs.

**A** Quality Control **B** Distributed Computation of Allele Frequencies

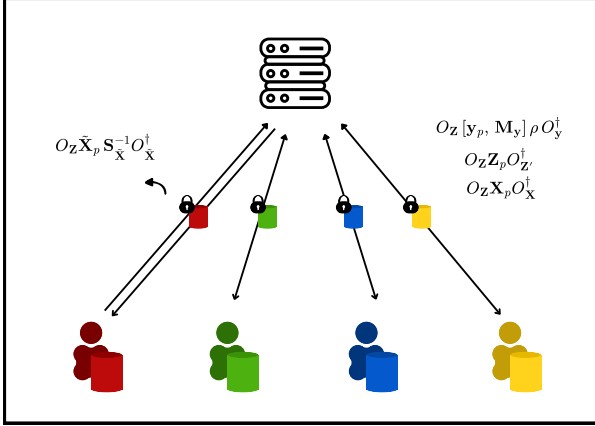

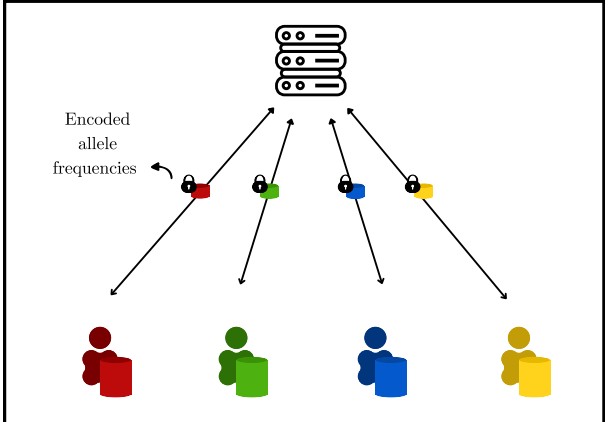

**C** Distributed Projection of Covariates and Standardization **D** Distributed Alternating Direction Method of Multipliers

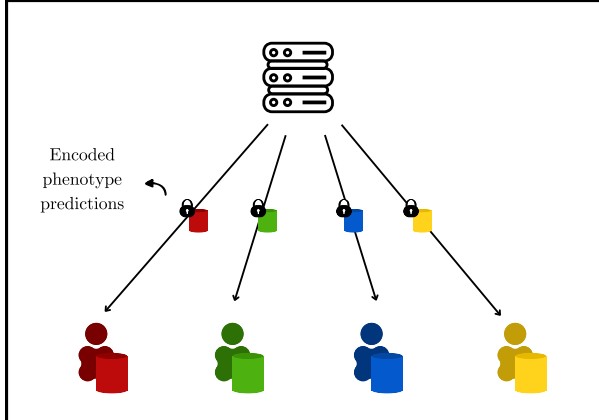

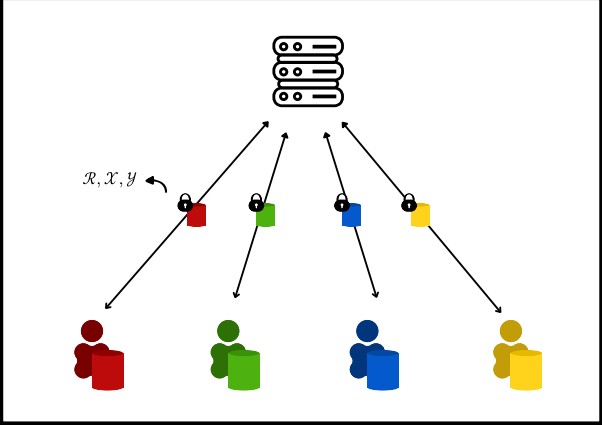

**E** Conjugate Gradient Descent **F** Distributed Single SNP Assoication Testing

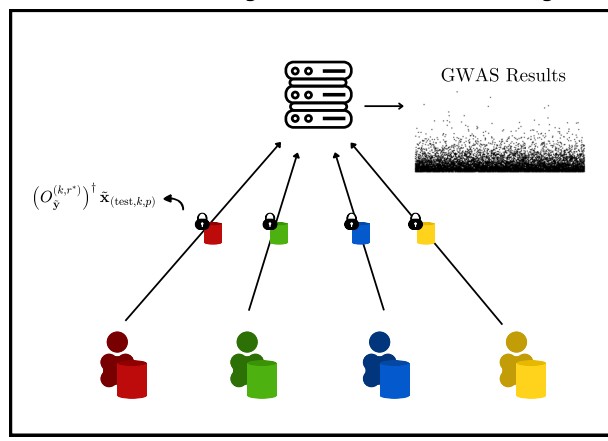

**Fig. 8 | Step-by-step illustration of PP-GWAS. A** Quality control and initialization: a common random seed is generated and securely shared among the computational nodes. **B** Allele-frequency estimation: with server coordination, nodes compute allele frequencies as in Eq. (18). **C** Covariate projection: nodes and server remove covariate effects as described in Eq. (20). **D** Level 0 model fitting: nodes transmit the aggregated quantities in Box 1 to enable distributed ADMM ridge regression. **E** Level 1 model fitting: the server performs ridge regression via conjugate gradient descent (CGD) on the ADMM outputs. **F** Single-SNP testing: nodes provide the quantities in Box 3; the server computes, for each SNP, a $\chi^2$ statistic (df = 1) and the corresponding two-sided $p$-value. At no point does the server access raw genotypes or phenotypes; only obfuscated intermediate values are exchanged.

optimizing the utilization of the kinship matrix, for instance, through matrix factorization methods.

REGENIE[21] employs a stacked ridge regression strategy and achieves an accuracy comparable to established tools such as BOLT-LMM[39], fastGWA[45], SAIGE[46], and FaST-LMM[47]. Since REGENIE is more friendly to distributed datasets, SF-GWAS[18] employed methods from MHE and MPC to build upon the algorithm. We similarly work with REGENIE in a distributed setting.

REGENIE executes its analysis in two phases. The initial phase involves a regression of the contributions from $\tilde{\mathbf{X}}$ out of $\tilde{\mathbf{y}}$, followed by

fitting $\beta_{\text{test}}$ on these adjusted residuals to ascertain associations. To mitigate the computational demands posed by the extensive genome-wide matrix $\tilde{\mathbf{X}}$, REGENIE implements a stacked ridge regression, executed in two distinct phases: Level 0 and Level 1. This approach significantly enhances computational efficiency and adaptability for large-scale genomic datasets, marking a notable progression in the field of genetic association studies.

At *Level* 0, the genotype matrix $\mathbf{X}$ is partitioned into $B$ vertical blocks, denoted as $\tilde{\mathbf{X}} = \left(\tilde{\mathbf{X}}^1, \ldots, \tilde{\mathbf{X}}^B\right)$. A set of $R$ distinct ridge parameters $\{\lambda_1, \ldots, \lambda_R\}$ are then chosen, where

$$\lambda_r := \frac{M(1 - h_r^2)}{h_r^2}, \quad h_r := \frac{0.01(R-1) + 0.98(r-1)}{R-1}. \quad (5)$$

Here, $M$ is the number of SNPs in the study. Consequently, $R$ ridge estimators are computed for each block:

$$\hat{\boldsymbol{\beta}}_{\lambda_r}^b = \left(\tilde{\mathbf{X}}^{b\top} \tilde{\mathbf{X}}^b + \lambda_r \mathbf{I}_{n \times n}\right)^{-1} \tilde{\mathbf{X}}^{b\top} \tilde{\mathbf{y}}, \quad (6)$$

$$\hat{\mathbf{y}}^{(b,r)} := \tilde{\mathbf{X}}^b \hat{\boldsymbol{\beta}}_{\lambda_r}^b. \quad (7)$$

These intermediate predictors $\hat{\mathbf{y}}^{(b,r)}$ for each block are then aggregated into a global feature matrix: $\mathbf{W}^b := \left(\hat{\mathbf{y}}^{(b,1)}, \ldots, \hat{\mathbf{y}}^{(b,R)}\right)$, $\mathbf{W} := \left(\mathbf{W}^1, \ldots, \mathbf{W}^B\right)$. This is implemented in a $k$-fold cross-validation framework, and hence we denote the $k$th folds of data as $\tilde{\mathbf{X}}_{(\text{LOCO},k)}^b$ and $\tilde{\mathbf{y}}_{(k)}$, and the data without the $k$th fold as $\tilde{\mathbf{X}}_{(\text{LOCO},k-1)}^b$ and $\tilde{\mathbf{y}}_{(k-1)}$. Hence, we have

$$\hat{\boldsymbol{\beta}}_{(\lambda_r, k-1)}^b = \left(\tilde{\mathbf{X}}_{(\text{LOCO},k-1)}^{b\top} \tilde{\mathbf{X}}_{(\text{LOCO},k-1)}^b + \lambda_r \mathbf{I}_{n \times n}\right)^{-1} \tilde{\mathbf{X}}_{(\text{LOCO},k-1)}^{b\top} \tilde{\mathbf{y}}_{(k-1)}, \quad (8)$$

$$\hat{\mathbf{y}}_{(\text{LOCO},k)}^{(b,r)} := \tilde{\mathbf{X}}_{(\text{LOCO},k)}^b \hat{\boldsymbol{\beta}}_{(\lambda_r, k-1)}^b. \quad (9)$$

At *Level 1*, a subsequent round of ridge regression is conducted on the intermediate feature matrix of size $N \times BR$, using $R$ parameters

$$\{\omega_1, \ldots, \omega_R\} = \left\{(BR/M)\lambda_1, \ldots, (BR/M)\lambda_R\right\}. \quad (10)$$

The ridge estimators are thus $\hat{\boldsymbol{\eta}}_r = \left(\mathbf{W}^\top \mathbf{W} + \omega_r \mathbf{I}_{BR \times BR}\right)^{-1} \mathbf{W}^\top \tilde{\mathbf{y}}$. The optimal ridge parameter $r^*$ is selected by minimizing the residual sum of squares:

$$r^* = \arg \min_r \|\tilde{\mathbf{y}} - \mathbf{W} \hat{\boldsymbol{\eta}}_r\|^2. \quad (11)$$

Phenotype predictions by the stacked regression model are defined as $\hat{\mathbf{y}} = \mathbf{W} \hat{\boldsymbol{\eta}}_{r^*}$. Notably, these two levels of ridge regression are implemented within a $k$-fold cross-validation framework. The predictions for the $k$th fold $\hat{\mathbf{y}}_k$ are aggregated, where

$$\hat{\mathbf{y}}_k := \mathbf{W}_k \hat{\boldsymbol{\eta}}_{(k-1, r^*)}, \quad (12)$$

$$\hat{\boldsymbol{\eta}}_{(k-1, r)} := \left(\mathbf{W}_{k-1}^\top \mathbf{W}_{k-1} + \omega_r \mathbf{I}_{BR \times BR}\right)^{-1} \mathbf{W}_{k-1}^\top \tilde{\mathbf{y}}_{k-1}, \quad (13)$$

$$r^* = \arg \min_r \sum_{k=1}^K \|\tilde{\mathbf{y}}_k - \mathbf{W}_k \hat{\boldsymbol{\eta}}_{(k-1, r)}\|^2. \quad (14)$$

The global predictor $\hat{\mathbf{y}} := \sum_{k=1}^K \hat{\mathbf{y}}_k$ facilitates the calculation of the associated $\chi^2$ statistic with one degree of freedom for the variant being tested:

$$\chi^2 = \frac{\left(\tilde{\mathbf{x}}_{\text{test}}^\top (\tilde{\mathbf{y}} - \hat{\mathbf{y}})\right)^2}{\hat{\sigma}_e^2 \left(\tilde{\mathbf{x}}_{\text{test}}^\top \tilde{\mathbf{x}}_{\text{test}}\right)}, \quad \hat{\sigma}_e^2 := \frac{\|\tilde{\mathbf{y}} - \hat{\mathbf{y}}\|_2^2}{(N - C)}. \quad (15)$$

The SNPs that have a $\chi^2$ value above a significant threshold are taken to be associated with the phenotype. The exact threshold depends on the study[48], with a conventional threshold being a $p$-value of $5 \times 10^{-8}$.

### Randomized encoding

Randomized encoding is central to our approach for computing a function's outcome while masking its underlying inputs. Formally, given a function

$$f : \mathcal{X} \to \mathcal{Y}, \quad (16)$$

a randomized encoding of $f$ is defined by two components:
- A randomized function $\hat{f} : \mathcal{X} \times \mathcal{R} \to \hat{\mathcal{Y}}$ where $\mathcal{R}$ represents the randomness space.
- A deterministic decoder $\text{Dec} : \hat{\mathcal{Y}} \to \mathcal{Y}$.

A randomized encoding of $f$ then satisfies:

$$\text{Dec}\left(\hat{f}(x; r)\right) = f(x) \quad (17)$$

with high probability, yet $\hat{f}(x; r)$ reveals no more information about $x$ than $f(x)$ does. In other words, $\hat{f}$ injects structured noise $r$ that conceals the input $x$, while still allowing a valid output $f(x)$ to be recovered by the decoder. Specific instances of this concept can preserve additional relationships (such as dot products) if required by tasks. Having introduced RE, we now describe the overall PP-GWAS protocol, beginning with a distributed quality control step that leverages an addition-based randomized encoding scheme.

### Quality control

In our protocol, the initial stage involves rigorous quality control (QC) checks on the genetic data. This is crucial to ensure the data's integrity and reliability, which are foundational for the accuracy of any subsequent analyses. We adhere to stringent criteria for these checks: a missing rate below 0.1, a minor allele frequency (MAF) above 0.05, and a Hardy–Weinberg equilibrium (HWE) chi-squared test statistic threshold of 23.928. These thresholds are aligned with established GWAS standards, allowing us to filter single-nucleotide polymorphisms (SNPs) effectively. Consistent with existing policies, for instance, by the National Institutes for Health (NIH)[49], our process includes sharing the total counts of reference homozygous, heterozygous, and alternate homozygous alleles for each SNP with each participating node, a practice also mirrored in SF-GWAS. To preserve data confidentiality during the QC phase (Fig. 8A and B) in our distributed environment, since we only need to sum the total counts amongst all nodes, we implement simple addition-based randomized encoding in a server-assisted manner. To compute the sum $f(x) = \sum_{i=1}^P x_i$, party $i$ in possession of $r_i$ and $\sum_i^P r_i$ generated using the shared seed, sends to the server $\hat{f}(x_i; r_i) = x_i + r_i$ which the server sums as $\hat{f}(x; r) = \sum_{i=1}^P \hat{f}(x_i; r_i)$ and returns to all the nodes. They then remove $\sum_i^P r_i$ to obtain

$$\text{Dec}\left(\hat{f}(x, r)\right) = \sum_{i=1}^P \hat{f}(x_i, r_i) - \sum_{i=1}^P r_i = \sum_{i=1}^P x_i. \quad (18)$$

PP-GWAS does not necessitate traditional joint genotyping with centralized data[50]. For common-variant single-SNP association on well-imputed or high-coverage datasets, modern variant-

**BOX 1**

# Distributed ADMM algorithm for Level 0 ridge regression

**Input:** Each node $p$ in $\{1, ..., P\}$ knows matrices $\tilde{\mathbf{X}}^b_{(p,k)}$, $\tilde{\mathbf{X}}^b_{(p,k-1)}$, column vector $\tilde{\mathbf{y}}_{(p,k-1)}$, learning rate $\ell$, and the number of ridge regression parameters $R$. The server requires learning rate $\ell$, ridge regression parameters $\{\lambda_1, ..., \lambda_R\}$, and number of iterations $n$.

 **Output:** Obfuscated Level 0 predictions for block $b$, regression parameter $\lambda_r$, and fold $k$ given by $O^{(k,r)}_{\hat{\mathbf{y}}} \hat{\mathbf{y}}^{(b,r)}_k$.

1. Each node, using a shared seed, prepares a non-zero constant $k_{\tilde{\mathbf{y}}}$, rectangular matrices $O^{(k,r)}_{\hat{\mathbf{y}}}$, $O^{(k,b,r)}_{\tilde{\mathbf{X}}}$ for all $k$, $b$, $r$, ensuring

$$\mathbb{E}\left[(O^{(k,r)}_{\hat{\mathbf{y}}})^{\top} O^{(k,r)}_{\hat{\mathbf{y}}}\right] = \mathbf{I}_N, \text{ and } \mathbb{E}\left[(O^{(k,b,r)}_{\tilde{\mathbf{X}}})^{\top} O^{(k,b,r)}_{\tilde{\mathbf{X}}}\right] = \mathbf{I}_{N_b}.$$

2. For each combination of $k, b, r$:

    (a) Server initializes $\mathcal{X}^{(0)}$, $\mathcal{Y}^{(0)}$, $\mathcal{Z}^{(0)}$ to 0.

    (b) Nodes compute and share with the server:

    - $\mathcal{R}^{(p,k,b,r)} := O^{(k,b,r)}_{\tilde{\mathbf{X}}}\left[(\tilde{\mathbf{X}}^b_{(p,k-1)})^{\top} \tilde{\mathbf{X}}^b_{(p,k-1)} + \ell\,\mathbf{I}_{N_b}\right](O^{(k,b,r)}_{\tilde{\mathbf{X}}})^{\dagger}$,

    - $\mathcal{X}^{(p,k,b,r)} := \frac{1}{k_{\tilde{\mathbf{y}}}} (O^{(k,r)}_{\hat{\mathbf{y}}})^{\dagger} \tilde{\mathbf{X}}^b_{(p,k)} O^{(k,b,r)}_{\tilde{\mathbf{X}}}$,

    - $\mathcal{Y}^{(p,k,r)} := O^{(k,r)}_{\hat{\mathbf{y}}} \tilde{\mathbf{y}}_{(k)}$,

    (c) Server computes:

    - $\mathcal{X}^{(1)}_p = \mathcal{R}^{(p,k,b,r)}\left(\sum_{\hat{k} \neq k} \mathcal{X}^{(p,\hat{k},b,r)}\right)^{\dagger} \mathcal{Y}^{(p,k,r)}$,

    - $\mathcal{X}^{(1)} = \sum_p \mathcal{X}^{(1)}_p / P$,

    - $\mathcal{Z}^{(1)} = \ell \mathcal{X}^{(1)} / \lambda_r$,

    - $\mathcal{Y}^{(1)}_p = \ell\left(\mathcal{X}^{(1)}_p - \mathcal{Z}^{(1)}\right)$,

    - $\mathcal{Y}^{(1)} = \sum_p \mathcal{Y}^{(1)}_p$.

    (d) For $i$ in $\{1, ..., n-1\}$:
    - Server updates as follows:

$$\mathcal{X}^{(i+1)}_p = \mathcal{R}^{(p,k,b,r)}\left(\left(\sum_{\hat{k} \neq k} \mathcal{X}^{(p,\hat{k},b,r)}\right)^{\dagger}\left(\sum_{\hat{k} \neq k} \mathcal{Y}^{(p,\hat{k},r)}\right) + \ell\,\mathcal{Z}^{(i)} - \mathcal{Y}^{(i)}_p\right),$$

$$\mathcal{X}^{(i+1)} = \sum_p \mathcal{X}^{(i+1)}_p / P,$$

$$\mathcal{Z}^{(i+1)} = \ell \mathcal{X}^{(i+1)} + \mathcal{Y}^{(i)} / \lambda_r,$$

$$\mathcal{Y}^{(i+1)}_p = \mathcal{Y}^{(i)} + \ell(\mathcal{X}^{(i+1)}_p - \mathcal{Z}^{(i+1)}),$$

$$\mathcal{Y}^{(i+1)} = \sum_p \mathcal{Y}^{(i+1)}_p.$$

    (e) Server computes $\mathcal{X}^{(p,k,b,r)} \mathcal{Z}^{(n)} = O^{(k,r)}_{\hat{\mathbf{y}}} \hat{\mathbf{y}}^{(b,r)}_k$.

 **Return:** $O^{(k,r)}_{\hat{\mathbf{y}}} \hat{\mathbf{y}}^{(b,r)}_k$.

calling and imputation pipelines achieve high accuracy, limiting the benefits of joint genotyping[51–53]. Second, our globally performed QC retains variants present in any participating site, so rare variants that might otherwise be discarded by site-specific QC are preserved. This realizes a principal benefit of joint genotyping where rare SNPs absent at a cohort will be "rescued"[51]. When using our method in collaborative settings, in the absence of joint genotyping, data harmonization is required to identify a common set of SNPs across sites. This can be achieved by sharing only non-private information, such as genomic positions, reference, alternate alleles, strand information, and when available, the rsID, so an aggregator can build a common SNP list without exposing individual-level data. These steps, together with covariate projection, mitigate technical artefacts, but do not eliminate all effects of technical heterogeneity. We note that a privacy-preserving distributed joint-genotyping layer could further reduce such heterogeneity without centralizing raw data and is complementary to PP-GWAS, but outside the scope of this work.

**Distributed projection of covariates and standardizing**

In our framework, the genomic information $\mathbf{X}$, covariate information $\mathbf{Z}$, and phenotype information $y$ are horizontally partitioned across $P$ computational nodes, with each node $p$ holding $\mathbf{X}_p$, $\mathbf{Z}_p$, and $\mathbf{y}_p$. Each node maintains a count of the total number of samples added to the study prior to their inclusion and the overall sample count. This information is conveyed through a sequential onboarding process. At the outset of the study, all the nodes establish a shared secret key using established cryptographic techniques, $k_{\text{seed}}$, unknown to the server. This secret key serves as the seed for generating subsequent shared keys. We denote that we have $N$ samples in total, $M$ SNPs, $C$ covariates, and $B$ blocks, which can be inferred by the server.

Subsequently, we standardize the genomic matrix $\mathbf{X}$ and phenotype matrix $\mathbf{y}$, and project out covariate information $\mathbf{Z}$ in the same computation (Fig. 8 C). We do this by appending $\mathbf{Z}$ with a column of ones to mean-center $\mathbf{X}$ and $\mathbf{y}$. We shall denote the updated covariate matrix as $\mathbf{Z}_1$. We also pre-compute the standard deviation matrix $\mathbf{S_X}$ of $\mathbf{X}$ and the standard deviation $s_\mathbf{y}$ of phenotype information $\mathbf{y}$ using the same addition-based randomized encoding approach as before, since

---

**BOX 2**

# CGD algorithm for Level 1 ridge regression

---

**Input:** The server requires a number of iterations $n$, ridge regression parameters $\omega_1, \ldots, \omega_R$, pre-received $\mathcal{Y}^{(p, k, r)}$ from the level 0 computation, and the precomputed $O_{\tilde{\mathbf{y}}}^{(k, r)} \hat{\mathbf{y}}_k^{(b, r)}$ for all $k, b, r$.

**Output:** Obfuscated Level 1 predictions for regression parameter $\omega_r$ and fold $k$ given by $O_{\tilde{\mathbf{y}}}^{(k, r)}[\hat{\mathbf{y}}_k^{(1, r)}, \ldots, \hat{\mathbf{y}}_k^{(B, r)}] \hat{\boldsymbol{\eta}}_{(k-1, r)}$.

1. For each combination of $k, r$:
   (a) Server initializes $\mathcal{X}^{(0)} = 0$.

   (b) Server initializes $\mathcal{Y}^{(0)}$, $\mathcal{Z}^{(0)} = \sum_{\hat{k} \neq k} \left( \left( O_{\tilde{\mathbf{y}}}^{(\hat{k}, r)} \hat{\mathbf{y}}_{\hat{k}}^{(b, r)} \right)^{\dagger} \left( \sum_{p=1}^{P} \mathcal{Y}^{(p, \hat{k}, r)} \right) \right)$..

   (c) Server precomputes $\mathcal{W} = \sum_{\hat{k} \neq k} \left( O_{\tilde{\mathbf{y}}}^{(k, r)} \hat{\mathbf{y}}_{\hat{k}}^{(b, r)} \right)^{\dagger} \left( O_{\tilde{\mathbf{y}}}^{(\hat{k}, r)} \hat{\mathbf{y}}_{\hat{k}}^{(b, r)} \right)$.

   (d) For $i$ in $0, \ldots, n-1$:
   - $\alpha = \mathcal{W} \mathcal{Y}^{(i)}$, ,
   - $\alpha + \omega_r \mathcal{Y}^{(i)}$,
   - $\gamma = (\mathcal{Z}^{(i)})^{\dagger} \mathcal{Z}^{(i)} / (\mathcal{Y}^{(i)})^{\dagger} \alpha$,
   - $\mathcal{X}^{(i+1)} = \mathcal{X}^{(i)} + \gamma \mathcal{Y}^{(i)}$,
   - $\mathcal{Z}^{(i+1)} = \mathcal{Z}^{(i)} - \gamma \alpha$,
   - $\delta = (\mathcal{Z}^{(i+1)})^{\dagger} \mathcal{Z}^{(i+1)} / (\mathcal{Z}^{(i)})^{\dagger} \mathcal{Z}^{(i)}$,
   - $\mathcal{Y}^{(i+1)} = \mathcal{Z}^{(i+1)} + \delta \mathcal{Y}^{(i)}$.

   (e) Server computes

$$r^{*} := \sum_k \mathrm{gramin}_r \, \| \sum_p \bar{\mathcal{Y}}^{(p, k, r)} - O_{\tilde{\mathbf{y}}}^{(k, r)}[\hat{\mathbf{y}}_k^{(1, r)}, \ldots, \hat{\mathbf{y}}_k^{(B, r)}] \hat{\boldsymbol{\eta}}_{(k-1, r)} \|_2^2.$$

**Return:**

$O_{\tilde{\mathbf{y}}}^{(k, r^{*})}[\hat{\mathbf{y}}_k^{(1, r^{*})}, \ldots, \hat{\mathbf{y}}_k^{(B, r^{*})}] \mathcal{X}^{(n)} = O_{\tilde{\mathbf{y}}}^{(k, r^{*})}[\hat{\mathbf{y}}_k^{(1, r^{*})}, \ldots, \hat{\mathbf{y}}_k^{(B, r^{*})}] \hat{\boldsymbol{\eta}}_{(k-1, r^{*})}$.

---

we only need to sum up relevant allele counts from each node. We can then project out covariates in a single computation since we know that

$$\tilde{\mathbf{X}} := \left( \mathbf{I}_N - \mathbf{Z}(\mathbf{Z}^{\top}\mathbf{Z})^{-1}\mathbf{Z}^{\top} \right) \mathbf{X}_S = \left( \mathbf{I}_N - \mathbf{Z}_1(\mathbf{Z}_1^{\top}\mathbf{Z}_1)^{-1}\mathbf{Z}_1^{\top} \right) \mathbf{X} \, \mathbf{S}_{\mathbf{X}}. \quad (19)$$

Here $\mathbf{X}_S$ denotes $\mathbf{X}$ after standardization. We do this since covariate projection inherently corrects for site-level batch effects by adjusting for technical covariates in the model. In settings where cohorts differ by sequencing platform, or variant-calling pipeline, each node can encode platform, pipeline, batch indicators as covariates to correct for potential artefacts as is the standard across various studies[21,39–44].

To perform Eq. (19) in a distributed and privacy-preserving manner, we treat the computation as a randomized encoding task, i.e, $f(\mathbf{Z}, \mathbf{X}) = \tilde{\mathbf{X}}$. We adopt methods based on randomized projection from[24,29,30,54], where we achieve data obfuscation as described below. We first construct rectangular matrices $O_{\mathbf{X}}, O_{\mathbf{y}}, O_{\mathbf{Z}}$ and $O_{\mathbf{Z}'}$ that satisfy $\mathbb{E}\left[ O_{\mathbf{X}}^{\dagger} O_{\mathbf{X}} \right] = \mathbb{E}\left[ O_{\mathbf{y}}^{\dagger} O_{\mathbf{y}} \right] = \mathbb{E}\left[ O_{\mathbf{Z}}^{\dagger} O_{\mathbf{Z}} \right] = \mathbb{E}\left[ O_{\mathbf{Z}'}^{\dagger} O_{\mathbf{Z}'} \right] = \mathbf{I}$. Each node $p$ prepares encoded data in the form of $O_{\mathbf{Z}}\mathbf{Z}_p O_{\mathbf{Z}'}^{\dagger}, O_{\mathbf{Z}}\mathbf{X}_p O_{\mathbf{X}}^{\dagger}$, and $O_{\mathbf{Z}}[\mathbf{y}_p, \mathbf{M}_{\mathbf{y}}]\rho O_{\mathbf{y}}^{\dagger}$ and sends them to the server. Here $\mathbf{M}_{\mathbf{y}}$ is a random matrix with $N$ rows, and $\rho$ a permutation matrix. We note that all the random matrices here are prepared with the help of the shared seed $k_{\mathrm{seed}}$. The server then computes for each node,

$$O_{\mathbf{Z}}\tilde{\mathbf{X}}_p \, \mathbf{S}_{\tilde{\mathbf{X}}}^{-1} O_{\tilde{\mathbf{X}}}^{\dagger} = O_{\mathbf{Z}}\mathbf{X}_p O_{\mathbf{X}}^{\dagger} - O_{\mathbf{Z}}\mathbf{Z}_p O_{\mathbf{Z}'}^{\dagger} \left( \mathbf{T}^{\dagger}\mathbf{T} \right)^{-1} \mathbf{T}^{\dagger} \sum_{p=1}^{P} \left( O_{\mathbf{Z}}\mathbf{X}_p O_{\mathbf{X}}^{\dagger} \right), \quad (20)$$

$$\mathbf{T} := O_{\mathbf{Z}}\mathbf{Z}O_{\mathbf{Z}'} = \sum_{p=1}^{P} \left( O_{\mathbf{Z}}\mathbf{Z}_p O_{\mathbf{Z}'}^{\dagger} \right), \quad (21)$$

and sends these to the appropriate nodes. The nodes can then compute $\mathbb{E}[\tilde{\mathbf{X}}_p] = O_{\mathbf{Z}}^{\dagger}(O_{\mathbf{Z}}\tilde{\mathbf{X}}_p \mathbf{S}_{\tilde{\mathbf{X}}}^{-1} O_{\tilde{\mathbf{X}}}^{\dagger}) O_{\tilde{\mathbf{X}}} \mathbf{S}_{\mathbf{X}}$. Analogously, the nodes compute $\mathbb{E}[[\tilde{\mathbf{y}}_p, \mathbf{M}_{\mathbf{y}}]\rho]$ and retrieve $\mathbb{E}[\tilde{\mathbf{y}}_p]$ by undoing the permutation. Hence, we have estimated our computation $f(\mathbf{Z}, \mathbf{X})$ with $\hat{f}(\mathbf{Z}, \mathbf{X}; O_{\mathbf{Z}}, O_{\mathbf{X}})$ using $O_{\mathbf{Z}}$ and $O_{\mathbf{X}}$ that act as structured noise. Similarly, we have computed $f(\mathbf{Z}, \mathbf{y})$ with $\hat{f}(\mathbf{Z}, \mathbf{y}; O_{\mathbf{Z}}, O_{\mathbf{y}}, \mathbf{M}_{\mathbf{y}}, \rho)$.

**Level 0 ridge regression using distributed ADMM**
Next, we would like to perform the first level of ridge regression on the genotypes against the phenotypes, using $R$ parameters $(\lambda_1, \ldots, \lambda_R)$ given by Eq. (5) (Fig. 8D). We now estimate $\hat{\boldsymbol{\beta}}_{\lambda_r}^b$ for all blocks $b$ from Eq. (6). For this purpose, we adopt the distributed Alternate Direction Method of Multipliers[55] to jointly estimate the level 0 predictions. Note that on a centralized dataset, the ridge regression problem can be formulated as the following optimization problem for a given ridge parameter $\lambda_r$: $\hat{\boldsymbol{\beta}}_{\lambda_r}^b = \mathrm{gramin}_{\boldsymbol{\beta}} (\|\tilde{\mathbf{X}}^b \boldsymbol{\beta} - \bar{\mathbf{y}}\|_2^2 + \lambda_r \|\boldsymbol{\beta}\|_2^2)$. We introduce a variable $\flat$ to rewrite the equation as a constraint problem below.

$$\hat{\boldsymbol{\beta}}_{\lambda_r}^b = \arg \min_{\boldsymbol{\beta}, \mathbf{b}} \left( \|\tilde{\mathbf{X}}^b \boldsymbol{\beta} - \bar{\mathbf{y}}\|_2^2 + \lambda_r \|\mathbf{b}\|_2^2 \right), \quad \text{s.t.} \; \boldsymbol{\beta} - \mathbf{b} = 0. \quad (22)$$

Since the data in our setting is horizontally partitioned, we can rewrite Eq. (22) as follows, where we also horizontally partition $\beta$.

$$\hat{\boldsymbol{\beta}}_{\lambda_r}^b = \arg \min_{\{\boldsymbol{\beta}_p\}, \mathbf{b}} \left( \sum_{p=1}^{P} \|\tilde{\mathbf{X}}_p^b \boldsymbol{\beta}_p - \bar{\mathbf{y}}\|_2^2 + \lambda_r \|\mathbf{b}\|_2^2 \right), \quad \boldsymbol{\beta}_p - \mathbf{b} = 0 \; \forall p. \quad (23)$$

We detail our distributed approach to use randomized encoding to compute Eq. (23) in Box 1 below. The computational nodes use their

## BOX 3
# Distributed association testing

**Input:** The server requires pre-received $\bar{\mathcal{y}}^{(p,k,r)}$ from the level 0 computation and the precomputed $\mathcal{K}_k := O_{\bar{\mathbf{y}}}^{(k,r^*)} [\hat{\mathbf{y}}_k^{(1,r^*)}, \ldots, \hat{\mathbf{y}}_k^{(B,r^*)}] \hat{\boldsymbol{\eta}}_{(k-1,r^*)}$ from the level 1 computation for all $k$.

**Output:** $\chi^2$ value associated to SNP $\tilde{\mathbf{x}}_{\text{test}}$.

1. Nodes compute and share $\left(O_{\bar{\mathbf{y}}}^{(k,r^*)}\right)^{\dagger} \tilde{\mathbf{x}}_{(\text{test},k,p)}$ for all $k$.

2. Server computes

$$\chi^2_{\tilde{\mathbf{x}}_{\text{test}}} = \frac{\left[\sum_{k=1}^{K} \left(\sum_{p=1}^{P} \left(O_{\bar{\mathbf{y}}}^{(k,r^*)}\right)^{\dagger} \tilde{x}_{(\text{test},k,p)}\right)^{\dagger} \left(\sum_{p=1}^{P} \mathcal{y}^{(p,k,r^*)} - \mathcal{K}_k\right)\right]^2}{\hat{\sigma}^2 \sum_{k=1}^{K} \sum_{p=1}^{P} \left(\left(O_{\bar{\mathbf{y}}}^{(k,r^*)}\right)^{\dagger} \tilde{x}_{(\text{test},k,p)}\right)^{\dagger} \left(\left(O_{\bar{\mathbf{y}}}^{(k,r^*)}\right)^{\dagger} \tilde{x}_{(\text{test},k,p)}\right)}, \tag{24}$$

where

$$\hat{\sigma}^2 := \frac{1}{N-C} \sum_{k=1}^{K} \|\sum_{p=1}^{P} \mathcal{y}^{(p,k,r^*)} - \mathcal{K}_k\|_2^2. \tag{25}$$

**Return:** $\chi^2_{\tilde{\mathbf{x}}_{\text{test}}}$.

shared seed to consistently segregate their data into $B$ blocks. They also then use the seed to determine how they split their data vertically into $K$ folds, such that every node has some data in every fold. They then denote the $k$th fold as $\tilde{\mathbf{X}}_{(p,k)}^b$ and the data without the $k$th fold as $\tilde{\mathbf{X}}_{(p,k-1)}^b$. Similarly, they have $\tilde{\mathbf{y}}_{(p,k)}$ and $\tilde{\mathbf{y}}_{(p,k-1)}$.

In this distributed ADMM framework, each computational node independently updates its local estimate $\beta_p$ by minimizing its respective objective, while a central variable $\mathfrak{b}$ is iteratively updated to enforce consensus among the nodes. The method involves alternating updates of the local variables and dual variables, ensuring that the global constraint $\beta_p - \mathfrak{b} = 0$ is satisfied as the algorithm converges. In the algorithm, the local ADMM updates $\mathcal{x}_p^{(i)}$ correspond to the variables $\beta_p$ from Eq. (23), and the consensus variable $\mathfrak{b}$ is represented by $\mathcal{Z}^{(i)}$.

### Level 1 Ridge regression using CGD

Now, like before in the centralized formulation as in Eq. (7), we have reduced our problem to lower dimensionality. We then perform Conjugate Gradient descent (Fig. 8E), however, on the server's side on the obfuscated data, as described in Box 2 below. For this, the server prepares $R$ ridge regression parameters $(\omega_1, \ldots, \omega_R)$ given by Eq. (10). In this CGD framework, the variable $\mathcal{x}^{(i)}$ represents the current estimate of the lower-dimensional solution (analogous to the parameter vector in Eq. (7)), while $\mathcal{Z}^{(i)}$ and $\mathcal{y}^{(i)}$ correspond to the residual and conjugate direction vectors, respectively. These mappings ensure that the iterative updates converge to the optimal ridge regression solution on the obfuscated data.

### Distributed single SNP association testing

For the next stage of the analyses, the nodes engage in a one-off communication with the server, helping them retrieve the $\chi^2$ values associated with each SNP (Fig. 8F). This is outlined in Box 3 below. Note that the server sees the final $\chi^2$ values, but has no direct access to the underlying genotype or phenotype data. Furthermore, in case this also needs to be hidden, one can shuffle the ordering of SNPs in the study, preventing the server from linking specific $\chi^2$ statistics to identifiable SNP positions. The computational nodes can apply thresholds using standard criteria on these $p$-values locally.

### Privacy analysis

We now describe the privacy guarantees of our algorithm within an adversarial framework, comprising a subset of semi-honest computational nodes and/or a semi-honest non-colluding central server. We show that a corrupted participant is unable to extract any information about the data of other non-corrupt nodes, and similarly, a corrupted server is incapable of deducing any node-specific information. It is important to clarify that our analysis does not cover extreme data scenarios that automatically enable the prediction of block sizes. Our proof methodology aligns with the approaches documented in prior works[24,29,30,54,56].

**Theorem 1**. *PP-GWAS is secure against a semi-honest adversary who corrupts the central server.*

**Proof**. We define a semi-honest central server to be a third-party server that adheres to the prescribed protocol, but attempts to learn the private data. In PP-GWAS, the server receives encoded data

$$O_{\mathbf{Z}} \mathbf{Z}_p O_{\mathbf{Z}'}^{\top} \in \mathbb{C}^{(N+k_{\mathbf{Z}}) \times (C+k_{\mathbf{Z}'})}, \quad O_{\mathbf{Z}} \mathbf{X}_p O_{\mathbf{X}}^{\dagger} \in \mathbb{C}^{(N+k_{\mathbf{Z}}) \times (N+k_{\mathbf{X}})}, \tag{26}$$

$$O_{\mathbf{Z}} [\mathbf{y}_p, \mathbf{M}_{\mathbf{y}}] \rho\, O_{\mathbf{y}} \in \mathbb{C}^{(N+k_{\mathbf{Z}}) \times (1+k_{M_{\mathbf{y}}}+k_{\mathbf{y}})}, \quad \left(O_{\bar{\mathbf{y}}}^{(k,r^*)}\right)^{\dagger} \tilde{\mathbf{x}}_{(\text{test},k,p)} \in \mathbb{C}^{(N+k_{\bar{\mathbf{y}}}) \times 1}, \tag{27}$$

from each input node, where $N$ is the number of samples and $C$ is the number of covariates. The data that the central server then has access to includes

$$O_{\mathbf{Z}} \mathbf{Z}_p O_{\mathbf{Z}'}^{\top}, \quad O_{\mathbf{Z}} \mathbf{X}_p O_{\mathbf{X}}^{\dagger}, \tag{28}$$

$$O_{\mathbf{Z}} [\mathbf{y}_p, \mathbf{M}_{\mathbf{y}}] O_{\mathbf{y}}, \quad \left(O_{\bar{\mathbf{y}}}^{(k,r^*)}\right)^{\dagger} \tilde{\mathbf{x}}_{(\text{test},k,p)}. \tag{29}$$

It is evident that the block sizes are hidden from the central server. The first three quantities are obfuscated on both sides and provide sufficient privacy[24,29,30,54]. Now we show that $(O_{\bar{\mathbf{y}}}^{(k,r^*)})^{\dagger} \tilde{\mathbf{x}}_{(\text{test},k,p)}$ is not produced by a unique pair $(O_{\bar{\mathbf{y}}}^{(k,r^*)})^{\dagger}$ and $\tilde{\mathbf{x}}_{(\text{test},k,p)}$. For simplicity, we denote the quantities as $O_{\tilde{\mathbf{x}}}$ and $\tilde{\mathbf{x}}$. Given an orthogonal matrix

$U \in \mathbb{R}^{N \times N}$ with $U\mathbf{1} = \mathbf{1}$, $\check{\mathbf{x}}_p = U\tilde{\mathbf{x}}_p$ and $\check{O}_{\tilde{\mathbf{x}}} = O_{\tilde{\mathbf{x}}}U^\top$, we have $O_{\tilde{\mathbf{x}}}\tilde{\mathbf{x}}_p = \check{O}_{\tilde{\mathbf{x}}}\check{\mathbf{x}}_p$. Further, since $\tilde{\mathbf{x}}$ is standardized, so is $\check{\mathbf{x}}$, and hence the structure of $\check{\mathbf{x}}$ provides no additional information for the server.

**Theorem 2.** *PP-GWAS is secure against a proper subset of semi-honest nodes.*

**Proof.** We define a proper subset of corrupt nodes as any subset excluding at least one honest node. We assume corrupt nodes are semi-honest, meaning they follow the protocol but may attempt to learn additional information from the accessible data. Each node $p$, only receives the relevant $p$'th partitions, such as $\mathbb{E}[\tilde{\mathbf{X}}_p]$ and $\mathbb{E}[\tilde{\mathbf{y}}_p]$. Therefore if a proper subset of the corrupt nodes collude, they cannot access or infer information beyond their encoded partitions. Since our adversarial setting considers a non-colluding semi-honest central server, the server will not deviate from the protocol and share information pertaining to non-corrupt nodes with the corrupt nodes.

Therefore, we have shown that the data of non-corrupt nodes remains private and secure from a proper subset of semi-honest nodes, and/or a non-colluding semi-honest central server. Further, the central server does not at any point of the protocol learn the block sizes utilized.

### Reporting summary

Further information on research design is available in the Nature Portfolio Reporting Summary linked to this article.

### Data availability

The real-world datasets analyzed here are available via controlled access from the NCBI database of Genotypes and Phenotypes (dbGaP). The bladder cancer risk dataset ($n = 13,060$; phs000346.v2.p2 [https://www.ncbi.nlm.nih.gov/projects/gap/cgi-bin/study.cgi?study_id=phs000346.v2.p2]) and the age-related macular degeneration dataset ($n = 22,683$; phs001039.v1.p1[https://www.ncbi.nlm.nih.gov/projects/gap/cgi-bin/study.cgi?study_id=phs001039.v1.p1]) contain individual-level genomic and phenotypic information collected under informed consent and are therefore available only to qualified researchers under the Data Use Limitations specified in each dbGaP record. Access requests should be submitted through the dbGaP Authorized Access system, citing the accession numbers above and including an institutional Data Use Certification and, where applicable, IRB/ethics approval. Requests are reviewed by the appropriate NIH dbGaP Data Access Committee; the authors are not involved in approval decisions. Further details on the original study protocols, including participant recruitment and sample collection, are provided in the respective dbGaP records. Access requests are typically reviewed by the NIH Data Access Committee in about two weeks on average; if approved, dataset access is granted for one year and may be renewed. Synthetic data were generated using pysnptools. Instructions and scripts for generating these synthetic datasets are publicly available in our GitHub repository. No other custom datasets were generated for this study. Source data are provided with this paper for all figures and tables derived from testing on the synthetic data. Source data are provided with this paper.

### Code availability

Our code is available on GitHub at the following URL: https://github.com/mdppml/PP-GWAS[57].

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

## Acknowledgements

This research was supported by the German Federal Ministry of Education and Research (BMBF) (project 01ZZ2010; A.S., M.A., and N.P.) and, in part, by the PrivateAIM project (01ZZ2316D; M.A. and N.P.). We express our gratitude to Prof. Dr. Sven Nahnsen for providing access to the real-world datasets utilized in this study. Our gratitude also goes to Dr. Carl Kadie for their assistance in generating synthetic data. We acknowledge the usage of the Training Center for Machine Learning (TCML) cluster at the University of Tübingen. This work was further supported by the de.NBI Cloud within the German Network for Bioinformatics Infrastructure (de.NBI) and ELIXIR-DE (Forschungszentrum Jülich and W-de.NBI-001, W-de.NBI-004, W-de.NBI-008, W-de.NBI-010, W-de.NBI-013, W-de.NBI-014, W-de.NBI-016, W-de.NBI-022). We also thank Cem Ata Baykara, Larissa Reichart and Lukas Böhm for their help with debugging code errors. We acknowledge support from the Open Access Publication Fund of the University of Tübingen.

## Author contributions

A.B.U., A.S., and M.A. conceived the study. A.S and M.A. designed the study, with A.S. developing the theoretical framework. A.S. analyzed the data and conducted the experiments. A.H. contributed to the implementation of the socket architecture. A.S. wrote the manuscript, with feedback from A.B.U, A.H., M.A., and N.P. The manuscript was revised by A.S. and M.A., while M.A. supervised the project.

## Funding

## Competing interests

The authors declare no competing interests.

## Additional information

Arjhun Swaminathan or Mete Akgün.

**Peer review information** *Nature Communications* thanks Miran Kim and
the other, anonymous, reviewer(s) for their contribution to the peer
review of this work. A peer review file is available.

