## [Transparent Peer Review file · Nature Communications]

PP-GWAS: Privacy Preserving Multi-Site Genome-wide Association Studies

Corresponding Author: Mr Arjhun Swaminathan

Version 0:

Reviewer comments:

Reviewer #1

(Remarks to the Author)

This paper introduces Privacy-Preserving GWAS (PP-GWAS), a method for performing a collaborative GWAS across multiple parties. Unlike a regular GWAS, where all data needs to be aggregated in a central location, PP-GWAS allows each party to keep their own data private. The main benefit of PP-GWAS is to facilitate joint GWAS across datasets that cannot be brought together due to privacy laws or other data sharing restrictions.

PP-GWAS uses a similar framework to SF-GWAS, another privacy preserving GWAS approach. Both methods build off of the REGENIE method to combine local plaintext computation per party with global computation on encrypted data. SF-GWAS uses a combination of secure multiparty computation and homomorphic encryption which have formal security guarantees but can be computationally expensive. PP-GWAS uses random transformations of the input data and shares the transformed data with a trusted third party. The authors show that their approach provides several computation benefits compared to SF-GWAS including a faster runtime, less memory usage, and better scaling in number of parties, SNPs, and sample sizes.

However, PP-GWAS has several drawbacks to its approach that limit its potential contribution.

- There is no mention of “meta-analysis” in the whole paper. This is THE method used in the GWAS community to carry out analysis across studies. It is unclear what the benefits of using PP-GWAS compared to a meta-analysis of per party summary statistics. Meta-analysis preserves data privacy because each party analyzes their own data, and only shares per variant summary statistics. The authors should discuss the benefits and drawbacks of PP-GWAS versus a meta-analysis and include experiments comparing the two. I would be curious to see benchmarking experiments comparing PP-GWAS to a meta-analysis like the experiments the authors performed with SF-GWAS. The paper references 2 relatively recent papers (4 and 5) but meta-analysis in GWAS has a long history that goes back to around 2007. There would be much better papers to reference and also to give a sense that meta-analysis has been used successfully for a long time without major issues.
- The real datasets used are not large by current standards. Analyzing the whole of the UK Biobank dataset together with another large Biobank would be a very relevant problem to solve. Especially if these were stored and analyzed on different computing systems.
- You could say a lot more about what S-GWAS and SF-GWAS actually do aimed at someone who works in the GWAS community. As written it assume a lot of existing knowledge, or requires reading references in detail.
- The algorithms presented appear to be for quantitative traits only, but the real datasets analyzed are two binary traits: bladder cancer and AMD. While balanced binary traits are sometimes analyzed as quantitative traits, this is a step back compared to the SF-GWAS approach which implements logistic regression for binary traits. Consequently, the results in Figure 2 look more concordant than I would expect. Unless the authors ran REGENIE treating AMD and bladder cancer as quantitative traits, I would not expect the correlation in log₁₀ p-values to be so high. The authors should either: i) attempt to develop similar methods for binary traits, or ii) clearly address this limitation in the main text. The authors should also include details on the parameters used with REGENIE so readers can better understand their experiments.
- SF-GWAS also includes methods for computing genetic principal components that PP-GWAS does not. Even in an LMM framework, principal components can be useful for correcting population structure or technical artifacts like batch effects across multiple collaborating institutions.

Separately, I have a couple questions about the privacy analysis in Appendix C. Theorem 1 shows that the method is secure against a compromised central server. One component of the argument seems to be that the sample size is unknown to the

server. But sample sizes are often disclosed when presenting results, or even before an analysis takes place (e.g. for a power calculation). It's unreasonable to assume this number would remain private forever. If the sample size is known, can the input data be recovered?

A second component of the argument uses input data data is transformed by a rectangular matrix with a left inverse. The server sees the transformed data, but not the inverse. The authors frame the problem of approximating the input data as an unbalanced Procrustes problem, which they state is currently unsolved. But being unsolved is different than not solvable. Is it possible that there will be a solution in the future to the unbalanced Procrustes problem? Is the unbalanced Procrustes problem the only possibly approach to recovering the input data? If not, could some other technique be used to recover the input?

In addition to the comments above, I have several minor comments below:

- Please include line numbers and equation numbers in the manuscript. It's helpful to reference parts of the manuscript.
- It would be helpful if the authors included a separate Methods section with additional details of the experiments performed. The PP-GWAS algorithm is described well, but key details from other analyses are missing from the manuscript.
- It looks like PP-GWAS only computes p-values. Is it possible to compute effect sizes and standard errors?
- It wasn't clear to me until page 6 that the number of computational nodes was referred to as participants. Why not use just use "computational nodes". In GWAS literature you often see study participants used for individuals in a study.
- Page 3 : The sentence "This often morphs into a dynamic challenge of identifying potential vulnerabilities and attacks rather than proving robustness from the outset." What is it exactly that "morphs"?
- Page 3 : you write "Subsequently, another round of ridge regression is performed on these predictions in a stacked fashion, and the associated SNPs are individually tested". It would be more accurate to say "Subsequently, another round of ridge regression is performed on these predictions in a stacked fashion, to produce a polygenic risk score (PRS), that is then used as a covariate when testing SNPs individually across the genome. The PRS is slightly adapted for each chromosome, using the Leave One Chromosome Out (LOCO) scheme, to avoid loss of signal due to local linkage disequilibrium (LD)."

(Remarks on code availability)

Reviewer #2

(Remarks to the Author)

(Remarks on code availability)

Reviewer #3

(Remarks to the Author)

[Technical novelty] The method section is lacking a detailed methods section, so it is difficult to evaluate the novelty of the proposed techniques in this manuscript. The authors utilize randomized encoding to obfuscate the data, which was already introduced in the previous work [19].

Additionally, critical details about the preprocessing steps are missing - specifically, how to share a secret key and use this seed for generating the subsequent shared matrices (e.g., Step 1 in Algorithm 1, Step 1 in Algorithm 3).

[Comparison] It would be beneficial to include a comparison with other cryptographic methods. For example, Homomorphic Encryption requires data providers to access a third-party with high computational resources (e.g., cloud server), and the proposed method also relies on a third party to perform the computations.

[Experiments] A more realistic experimental setup should be considered. Specifically, it would be helpful to incorporate multiple servers in a LAN or WAN setting environment. This would provide a more accurate estimate of the communication costs between parties.

[Source code] The source code is available, but the README file is empty, and there is no guidance on how to run the source code.

(Remarks on code availability)

The source code is available, but the README file is empty, and there is no guidance on how to run the source code.

Reviewer #4

(Remarks to the Author)

This paper by Swaminathan et al. introduces a method for conducting GWAS while preserving data privacy across multiple sites. Although the approach addresses an important challenge in multi-site genomic research, I found the presentation of the methodology difficult to follow. Much of the technical detail is relegated to the appendices. This makes it hard for readers to grasp the core concepts without sifting through supplementary sections.

One key issue with the paper is the comparison with SF-GWAS. While the authors benchmark their method against SF-GWAS, the comparison seems unfair. SF-GWAS provides two approaches—one that accounts for population stratification and another that uses a linear mixed model (LMM). The proposed PP-GWAS method is specifically designed for LMM-based analyses, so it doesn't provide a full comparison with SF-GWAS's broader capabilities. A more balanced comparison, including other secure GWAS approaches, would give a clearer perspective on the relative strengths and weaknesses of the method.

Related to the above paragraph: Another critical gap in the manuscript is the absence of comparisons with other relevant secure and private GWAS methods. Important works that should be considered for benchmarking include Kockan, C., Zhu, K., Dokmai, N., et al., "Sketching algorithms for genomic data analysis and querying in a secure enclave," *Nat Methods*, 2020 and Li, Wentao et al., "Federated generalized linear mixed models for collaborative genome-wide association studies," *iScience*, 2023. I

The paper claims that its core contribution is in the domain of privacy. However, the privacy analysis appears only towards the end of the appendix. It is also written in a way that makes it less accessible for a wider audience. Given that privacy is a primary focus, the authors should place a greater emphasis on demonstrating that their method ensures privacy and security. Specifically, the paper would benefit from a more detailed and accessible explanation of how the method ensures that the encodings do not inadvertently reveal genetic information about participants.

Furthermore, while the paper acknowledges that the security of the encoding relies on avoiding reconstruction attacks, there is a lack of discussion of this matter. A thorough analysis of the potential risks posed by reconstruction attacks is important, especially in scenarios where adversaries might have access to additional data that could be used to reverse-engineer sensitive information. The authors should provide a deeper exploration of the likelihood and impact of such attacks, and how their method mitigates these risks.

The use of real-world data for validation is a strength of the study; however, the paper overlooks a key limitation of its approach. The datasets used are jointly genotyped, which simplifies the QC process. In real-world scenarios, data from different sites may be independently sequenced, which presents additional challenges for QC. The paper should address how the proposed method would handle variations in sequencing quality across sites, and how this might affect the robustness of the analyses.

The github is not helpful. There is no information on how to install and use the code. It is just bunch of folders with code in it. I was not able evaluate the code due to this lack of explanation.

Overall, while the paper tackles an important problem in the domain of privacy-preserving GWAS, it would benefit greatly from a clearer presentation of its methods, a more comprehensive privacy analysis, a fairer comparison with alternative approaches, and an exploration of real-world challenges.

(Remarks on code availability)

I am not able to run the code as there is no information on how to run it. There are also bunch of files in their results folder, which I hope does not contain any private information about their results with real-world data they used and it is only the simulated data.

Version 1:

Reviewer comments:

Reviewer #1

(Remarks to the Author)

The authors have addressed all my comments.

(Remarks on code availability)

Reviewer #2

(Remarks to the Author)

(Remarks on code availability)

Reviewer #3

(Remarks to the Author)

I think the authors have addressed my concerns in this revision.

I could not find detailed accuracy results on the synthetic dataset.

Additional clarification is needed on the following points:

- How do the participating nodes remove the aggregated noise (i.e., the summation of r_i in line 483)?
- How is the comparison with a specific threshold performed to determine whether a trait is above or below a given metric? Is this step executed locally using the total counts?

Furthermore, in the comparison with SF-GWAS, is the PCA step used in SF-GWAS excluded from the evaluation?

Minor comments:

It would be helpful to include the detailed sizes of the datasets when they are first introduced—for example, in the subsections “Synthetic Data Generation” and “Real Datasets.”

(Remarks on code availability)

Reviewer #4

(Remarks to the Author)

I believe the authors may have misunderstood my earlier point regarding joint genotyping. When I referred to the SNP data being “jointly genotyped,” I was not commenting on quality control procedures. I fully acknowledge that quality control will be conducted independently by each institution before the GWAS step. However, joint genotyping is a distinct process and should not be conflated with quality control.

Joint genotyping involves calling variants across multiple samples simultaneously, which improves consistency and reduces batch effects by leveraging population-level information to better distinguish true variants from sequencing or alignment artifacts. In a federated or distributed setting, joint genotyping is not feasible because raw sequencing data cannot be shared across institutions due to privacy and governance constraints. Consequently, the authors must work with SNP data that have been independently genotyped at each site.

This has important implications for their framework. Independent genotyping can introduce batch effects and inconsistencies in variant calls across sites, potentially affecting downstream analyses. Therefore, the authors should either (1) demonstrate that their method includes a mechanism to correct for this type of batch effect, or (2) provide evidence that the absence of joint genotyping does not substantially impact the accuracy or robustness of their framework.

(Remarks on code availability)

The code is not intuitive to review, as it requires setting up ports and configuring the framework. For reproducibility and easier code review, it would have been preferable to provide a single, streamlined workflow along with example toy data that can be run locally.

Version 2:

Reviewer comments:

Reviewer #3

(Remarks to the Author)

The authors have addressed all my comments.

(Remarks on code availability)

Reviewer #4

(Remarks to the Author)

I don't believe it's sufficient to address the joint genotyping issue by simply noting that prior work also did not consider it. Since I wasn't a reviewer on the earlier paper, I can't speak to the specifics of its review process or criteria.

More importantly, if SNPs are jointly genotyped, it raises the question of why a federated GWAS approach is needed—wouldn't that imply the genotypes are already centrally available? On the other hand, if SNPs are not jointly genotyped, could this introduce batch effects or other artifacts? Is there a way to quantify or correct for this? It would be helpful if the authors could include a simple experiment to illustrate whether and how this impacts the results. Do the findings change

meaningfully when joint genotyping is or isn't performed?

(Remarks on code availability)

I'm having trouble running the local_run. To start with, there's no README or documentation explaining what the components are or how to use them. I managed to make some partial progress using Google and ChatGPT, but the process was far from straightforward. There were no instructions on how to properly set up the conda environment required for the local run. Ultimately, I got stuck on an issue where the script failed to generate the neg_log_transfer.npy file, and I wasn't able to resolve it.

Version 3:

Reviewer comments:

Reviewer #4

(Remarks to the Author)

Thank you for your detailed response. However, I remain unconvinced that it addresses my key concern. Even the cited literature clearly shows that both the genotyping pipeline and sequencing platform strongly influence SNP distributions (see below for deeper dive). Therefore, you must either:

Empirically demonstrate that platform or pipeline effects are neutralized in your tool, or Include them as covariates, with a clear mechanism for correcting potential biases.

In this context, it's also important to clarify that joint genotyping is not merely the process of finding common variants across samples and imputing those absent in one dataset from another. Rather, its strength lies in performing variant calling within a unified framework—using the same pipeline across samples while explicitly considering sequencing platform as a covariate. This approach allows systematic biases introduced by different technologies to be properly accounted for, ensuring that variant calls are biologically meaningful rather than artifacts of machine or pipeline choice.

Additionally, I still cannot run the tool—please refer to the Code section for details on the errors I am encountering.

Below is relevant literature for context:

Chen et al. conducted a systematic comparison of 27 sequencer–variant caller combinations (including Illumina's NovaSeq, BGISEQ-500, MGISEQ-2000, using pipelines such as GATK-HC, Strelka2, and Samtools-VarScan2). This work found that although SNP calling F-scores remained high (>0.96 for WES, >0.975 for WGS), INDEL calling was far more variable (F-scores 0.71–0.93), highlighting diverging variant profiles across platform–pipeline combinations. Citation: Chen, J., Li, X., Zhong, H. et al. Systematic comparison of germline variant calling pipelines cross multiple next-generation sequencers. *Sci Rep* 9, 9345 (2019). <https://doi.org/10.1038/s41598-019-45835-3>

Foxx et al. compared germline variant callers across platforms including BGISEQ-500, MGISEQ-2000, and NovaSeq. They reported that DeepVariant consistently exhibited the highest accuracy, while GATK and Sentieon lagged behind—underlining the influence of both machine and software on variant outcomes. Citation: Foxx J, Tighe SW, Nicolet CM, Zook JM, Byrka-Bishop M, Clarke WE, Khayat MM, Mahmoud M, Laaguiby PK, Herbert ZT, Warner D, Grills GS, Jen J, Levy S, Xiang J, Alonso A, Zhao X, Zhang W, Teng F, Zhao Y, Lu H, Schroth GP, Narzisi G, Farmerie W, Sedlazeck FJ, Baldwin DA, Mason CE. Performance assessment of DNA sequencing platforms in the ABRF Next-Generation Sequencing Study. *Nat Biotechnol.* 2021 Sep;39(9):1129-1140. doi: 10.1038/s41587-021-01049-5. Epub 2021 Sep 9. Erratum in: *Nat Biotechnol.* 2021 Nov;39(11):1466. doi: 10.1038/s41587-021-01122-z. PMID: 34504351; PMCID: PMC8985210.

Abdelwahab et al. benchmarked conventional versus AI-based variant callers (e.g., BCFTools, GATK4, Platypus, DNAscope, DeepVariant) across Illumina, PacBio HiFi, and Oxford Nanopore data. They found that AI-based tools generally outperformed traditional ones, evidencing how platform error profiles and calling algorithms dictate variant call reliability. Citation: Abdelwahab, O., Belzile, F. & Torkamaneh, D. Performance analysis of conventional and AI-based variant callers using short and long reads. *BMC Bioinformatics* 24, 472 (2023). <https://doi.org/10.1186/s12859-023-05596-3>

(Remarks on code availability)

The code is still not running for me despite trying multiple approaches. To start, the environment file for the local run specifies the environment name as "ppgwas_test", but the README instructs users to activate "ppgwas-env". I adjusted it to "ppgwas_test", but the environment still lacked several essential dependencies—including jupyter, numpy, matplotlib, and pysnpTools. This suggests there's an underlying issue with the environment setup that I wasn't able to debug.

I attempted to work around this by manually installing the missing dependencies, but even then, I encountered persistent errors. I tried running the code in several ways (directly with Python and within a Jupyter notebook), and in both cases I received errors. I've copied below the error output I received from the notebook run.

Below it seems like the .sh file ran:

```
"Launching server on port 8000 (logs → logs/ppgwas_20250823_090756/server)
Spawning 4 clients (logs → logs/ppgwas_20250823_090756/clients)
```

All processes finished. Logs in logs/ppgwas_20250823_090756"

However, the error in the next block suggests that it is still lacking the log file

"-----

```
FileNotFoundError Traceback (most recent call last)
Cell In[8], line 5
      2 import matplotlib.pyplot as plt
      4 file_path = f"Data/N{N}_M{M}_C{C}_P{P}_B{B}/neg_log_transfer.npy"
-- 5 neg_log_p = np.load(file_path)
      6 positions = np.arange(len(neg_log_p))
      8 plt.figure(figsize=(10, 5))
```

```
File ~/miniconda3/envs/ppgwas_test/lib/python3.8/site-packages/numpy/lib/npymio.py:405, in load(file, mmap_mode,
allow_pickle, fix_imports, encoding, max_header_size)
    403 own_fid = False
    404 else:
 405 fid = stack.enter_context(open(os_fspath(file), "rb"))
    406 own_fid = True
    408 # Code to distinguish from NumPy binary files and pickles.
```

```
FileNotFoundError: [Errno 2] No such file or directory: 'Data/N10000_M15000_C10_P4_B2/neg_log_transfer.npy'"
```

Version 4:

Reviewer comments:

Reviewer #4

(Remarks to the Author)

I appreciate the author's detailed response to my concerns. However, I do not see these limitations reflected in the discussion section of the revised manuscript. As it stands, the discussion reads more like a promotion of the study and its advantages over other methods, rather than a balanced assessment. I strongly urge the authors to explicitly state the limitations that were acknowledged in their response and raised in my initial concerns. In particular, it should be made clear that:

- * There may be issues when genotypes are derived from non-joint genotyping across different sites.
- * The method would be most effective if implemented in the context of a privacy-preserving, federated joint genotyping framework.
- * It is unrealistic to expect clients to share genotypes, perform joint genotyping, and then return to a privacy-preserving GWAS model. Once genotypes are shared, the core rationale for federated GWAS is undermined.

Regarding the code, I must clarify that the problem was not with the kernel. I verified I was using the correct kernel. Upon detailed debugging, I found that the ppgwas.sh script attempts to activate a Conda environment (conda activate ppREGENIE) that does not exist in the local_run directory. Consequently, the environment fails to recognize the python command. Moreover, there is no environment file provided to create this ppREGENIE environment in the local_run folder. I attempted to substitute with ppgwas_test, but the job has been running for a very long time without clear indication of success, so it remains unclear whether this workaround is effective. (I also tried it by reducing number samples, SNPs and covariates to see if it was a runtime issue but it was still hanging and not finishing)

In my view, the repository requires a thorough cleanup and review by the authors themselves. A well-documented README, along with consistent environment files and validation of the code in a fresh setup (after removing all of their already installed environments), is essential to ensure reproducibility and usability for others.

(Remarks on code availability)

please see my comments above.

Version 5:

Reviewer comments:

Reviewer #4

(Remarks to the Author)

I thank the authors for addressing all my concerns. I have no remaining concerns.

I was able to run the code and produce results after making a small change on local_run/server.py

I changed the line "hostname = socket.gethostname(socket.gethostname())" with "hostname = socket.gethostname("localhost")" because otherwise my local DNS did not recognize it. I recommend making a note of this in the github in case others run into the same issue.

(Remarks on code availability)

Please see my comments to the authors.

Reviewer #1 (Remarks to the Author):

This paper introduces Privacy-Preserving GWAS (PP-GWAS), a method for performing a collaborative GWAS across multiple parties. Unlike a regular GWAS, where all data needs to be aggregated in a central location, PP-GWAS allows each party to keep their own data private. The main benefit of PP-GWAS is to facilitate joint GWAS across datasets that cannot be brought together due to privacy laws or other data sharing restrictions.

PP-GWAS uses a similar framework to SF-GWAS, another privacy preserving GWAS approach. Both methods build off the REGENIE method to combine local plaintext computation per party with global computation on encrypted data. SF-GWAS uses a combination of secure multiparty computation and homomorphic encryption which have formal security guarantees but can be computationally expensive. PP-GWAS uses random transformations of the input data and shares the transformed data with a trusted third party. The authors show that their approach provides several computation benefits compared to SF-GWAS including a faster runtime, less memory usage, and better scaling in number of parties, SNPs, and sample sizes.

We thank the reviewer for their thorough and insightful critique, which has substantially strengthened the clarity and scientific rigor of our work. We hope that the detailed, point-by-point responses below and the accompanying revisions fully address the concerns raised and are satisfactory.

However, PP-GWAS has several drawbacks to its approach that limit its potential contribution.

- **There is no mention of “meta-analysis” in the whole paper. This is THE method used in the GWAS community to carry out analysis across studies. It is unclear what the benefits of using PP-GWAS compared to a meta-analysis of per party summary statistics. Meta-analysis preserves data privacy because each party analyzes their own data, and only shares per variant summary statistics. The authors should discuss the benefits and drawbacks of PP-GWAS versus a meta-analysis and include experiments comparing the two. I would be curious to see benchmarking experiments comparing PP-GWAS to a meta-analysis like the experiments the authors performed with SF-GWAS. The paper references 2 relatively recent papers (4 and 5) but meta-analysis in GWAS has a long history that goes back to around 2007. There would be much better papers to reference and also to give a sense that meta-analysis has been used successfully for a long time without major issues.**

We thank the reviewer for highlighting the central role of meta-analysis in collaborative GWAS. In the revised manuscript we have (i) expanded the *Introduction* to provide a balanced discussion of meta-analysis, and (ii) added a new subsection, *Performance Evaluation against Meta-analysis* under the *Experiments* section. Here we benchmark PP-GWAS against standard meta-analysis (using PLINK) on both real and simulated datasets, varying the number of collaborating nodes. We observe that, while the two approaches exhibit comparable power when data are partitioned into a small number of computational nodes, the accuracy of meta-analysis degrades noticeably when the same dataset is split across an increasing number of nodes, whereas the performance of PP-GWAS is unaffected. Fig. 7 shows Manhattan plots from PP-GWAS and meta-analysis, illustrating the difference in association detection when data is split across six computational nodes.

- **The real datasets used are not large by current standards. Analyzing the whole of the UK Biobank dataset together with another large Biobank would be a very relevant problem to solve. Especially if these were stored and analyzed on different computing systems.**

We agree that demonstrating performance on truly large-scale cohorts is critical. Despite multiple attempts to obtain the exact dataset used by SF-GWAS for head-to-head comparison, we were unable to secure access. Hence we now

have updated the paper with experiments on simulated large scale data matching the size of the UK Biobank (\approx 275000 samples, 580,000 SNPs). For this purpose, we obtained the necessary compute allocations via the deNBI Cloud facility. Now, our section “Adaptability to Large Scale Data” has been updated. We have removed our interpolated results, and reported results when we ran a dataset the size of UK Biobank data. A direct head-to-head comparison with SF-GWAS was unfortunately infeasible: (i) the deNBI allocation we had access to limited us to four computational nodes and the server with varying RAM allocations and storage settings, and (ii) we were unable to obtain a working version of the SF-GWAS code despite repeated correspondence. We therefore reported their published runtimes and, using our empirically informed linear interpolation, provided an *estimated* runtime for PP-GWAS under identical hardware settings.

• You could say a lot more about what S-GWAS and SF-GWAS actually do aimed at someone who works in the GWAS community. As written it assume a lot of existing knowledge, or requires reading references in detail.

Thank you for your remark. We have now expanded our discussion on S-GWAS and SF-GWAS in our Introduction section for a better perspective.

• The algorithms presented appear to be for quantitative traits only, but the real datasets analyzed are two binary traits: bladder cancer and AMD. While balanced binary traits are sometimes analyzed as quantitative traits, this is a step back compared to the SF-GWAS approach which implements logistic regression for binary traits. Consequently, the results in Figure 2 look more concordant than I would expect. Unless the authors ran REGENIE treating AMD and bladder cancer as quantitative traits, I would not expect the correlation in log₁₀ p-values to be so high. The authors should either: i) attempt to develop similar methods for binary traits, or ii) clearly address this limitation in the main text. The authors should also include details on the parameters used with REGENIE so readers can better understand their experiments.

We acknowledge the confusion. PP-GWAS, like REGENIE, currently supports quantitative traits only. We treated the real datasets as quantitative, for example, by setting 1 to 1.00. The two real datasets (AMD and bladder cancer risk) were analysed using REGENIE’s --force-qt option, treating the phenotypes as quantitative (now stated explicitly) for appropriate comparison. We have revised the paper to clarify that the present work targets quantitative phenotypes. Extending PP-GWAS to logistic mixed-model association is a priority for future work.

• SF-GWAS also includes methods for computing genetic principal components that PP-GWAS does not. Even in an LMM framework, principal components can be useful for correcting population structure or technical artifacts like batch effects across multiple collaborating institutions.

We thank the reviewer for highlighting the role of PCA in SF-GWAS. As now detailed in the paper, SF-GWAS offers two workflows: one computes PCA followed by standard association testing, and the other implements a linear mixed-model (LMM) approach based on REGENIE. Our PP-GWAS pipeline focuses on the latter, performing REGENIE in a distributed, privacy-preserving manner using randomized encoding, and therefore does not currently include a PCA module. We recognize that secure PCA has potential to further strengthen LMM, and we plan to incorporate privacy-preserving PCA—leveraging randomized encoding methods as described in [1,2]—in future extensions of our framework. We would also emphasize that SF-GWAS primarily reports PCA-based results, and their LMM-based approach takes significantly more time, whereas our LMM-based approach outperforms their PCA-based approach.

Separately, I have a couple questions about the privacy analysis in Appendix C. Theorem 1 shows that the method is secure against a compromised central server. One component of the argument seems to be that the sample size is unknown to the server. But sample sizes are often disclosed when presenting results, or even before an analysis takes place (e.g. for a power calculation). It's unreasonable to assume this number would remain private forever. If the sample size is known, can the input data be recovered?

We thank the reviewer for this important observation. It is correct that, in practice, sample sizes are often disclosed in advance or reported alongside study results, and relying on their secrecy might prove impractical. To address this, we have extended the masking scheme by applying randomized encodings on both sides of Z , i.e., replacing the one-sided mask $O_Z Z$ with $O_Z Z O'_Z$. Hence the knowledge of the number of samples no longer risks recovery of Z . Because this modification occurs entirely during preprocessing when we project out covariates before more computationally expensive operations, it incurred no additional runtime to any of our experiments. Fig 3. C where we study the impact of the number of covariates on the runtime of PP-GWAS further supports this observation. We have updated the methods section to describe this change in our methods, and revised also the Privacy Analysis section accordingly.

A second component of the argument uses input data data is transformed by a rectangular matrix with a left inverse. The server sees the transformed data, but not the inverse. The authors frame the problem of approximating the input data as an unbalanced Procrustes problem, which they state is currently unsolved. But being unsolved is different than not solvable. Is it possible that there will be a solution in the future to the unbalanced Procrustes problem? Is the unbalanced Procrustes problem the only possibly approach to recovering the input data? If not, could some other technique be used to recover the input?

In tandem with our symmetric masking update above, we have removed reliance on the unbalanced Procrustes hardness assumption. The revised privacy argument now rests instead on the privacy guarantees offered by triple matrix masking [1,2,3,4].

In addition to the comments above, I have several minor comments below:

- **Please include line numbers and equation numbers in the manuscript. It's helpful to reference parts of the manuscript.**

We have now updated the manuscript accordingly.

- **It would be helpful if the authors included a separate Methods section with additional details of the experiments performed. The PP-GWAS algorithm is described well, but key details from other analyses are missing from the manuscript.**

Thank you for this comment. We have now restructured the manuscript as a whole to reflect the expectations of Nature Communications. Instead of having all information in the appendices, we have now included them accordingly in a detailed method's section.

- **It looks like PP-GWAS only computes p-values. Is it possible to compute effect sizes and standard errors?**

No. PP-GWAS as it stands now can only compute p-values of the association between individual SNPs and phenotypes.

- **It wasn't clear to me until page 6 that the number of computational nodes was referred to as participants. Why not use just use "computational nodes". In GWAS literature you often see study participants used for individuals in a study.**

Thank you for this comment. Our language is now consistent and we now refer to all data entities as computational nodes and the third party computational entity as the server throughout the study.

• **Page 3 : The sentence “This often morphs into a dynamic challenge of identifying potential vulnerabilities and attacks rather than proving robustness from the outset.” What is it exactly that “morphs”?**

We have now updated our language to be simpler and concise. The statement now reads “This translates into a dynamic challenge of identifying potential vulnerabilities and attacks rather than proving robustness from the outset.”

• **Page 3 : you write “Subsequently, another round of ridge regression is performed on these predictions in a stacked fashion, and the associated SNPs are individually tested”. It would be more accurate to say “Subsequently, another round of ridge regression is performed on these predictions in a stacked fashion, to produce a polygenic risk score (PRS), that is then used as a covariate when testing SNPs individually across the genome. The PRS is slightly adapted for each chromosome, using the Leave One Chromosome Out (LOCO) scheme, to avoid loss of signal due to local linkage disequilibrium (LD).”**

Thank you for pointing this out. We have updated our manuscript accordingly.

We would like to express our sincere gratitude to the reviewer for the time and effort invested in evaluating our work and providing feedback. Your suggestions and comments have been invaluable in strengthening the paper and in improving its scientific rigor. We appreciate the constructive feedback, and look forward to any further comments you may have.

[1] Hannemann, A., Ünal, A.B., Swaminathan, A., Buchmann, E., Akgün, M.: A privacy-preserving framework for collaborative machine learning with kernel methods

[2] Hannemann, A., Swaminathan, A., Ünal, A.B., Akgün, M.: Private, efficient and scalable kernel learning for medical image analysis

[3] Ding, A.A., Miao, G., Wu, S.S.: On the privacy and utility properties of triple matrix-masking.

[4] Liu, K., Kargupta, H., Ryan, J.: Random projection-based multiplicative data perturbation for privacy preserving distributed data mining.

List of changes:

Here, we provide with the list of changes made in our resubmission for the reference of both the editor and the reviewers. In our list of changes, we refer to the reviewers 1,2,3 and 4 as R1, R2, R3 and R4 respectively. Since we are unsure which reviewer R2 worked with on the paper, we are unable to mention them below.

1. Expanded Background and Comparative Methods (R1, R3, R4)

- a. The Introduction now includes a more comprehensive overview of meta-analysis, S-GWAS, and SF-GWAS.

2. Manuscript Restructuring (R1, R4)

- a. The paper has been fully restructured to align with Nature Communications' formatting standards.
- b. Content previously in the appendices, particularly in the Methods section, has been moved into the main text to improve readability.

- 3. Clarification on Quantitative Data Assumptions (R1)**
 - a. Throughout the manuscript, we now explicitly state that both PP-GWAS and SF-GWAS (built on LMM architectures) operate on quantitative phenotypes, consistent with the design of REGENIE which both methods are based on.
- 4. Updated SF-GWAS Benchmarking**
 - a. Since SF-GWAS was published very recently in Nature Genetics and had additional benchmarking, we now test the performance of PP-GWAS on datasets of varying covariate sizes, mirroring the new experiments reported in SF-GWAS (see Fig. 3C).
- 5. Large-Scale Performance Evaluation (R1)**
 - a. Additional benchmarks on UK Biobank-scale datasets have been performed utilizing more computational resources than was previously available with the help of deNBI Cloud (see Fig. 3E).
- 6. Network Condition Experiments (R3)**
 - a. We report the performance of PP-GWAS under diverse network conditions in the new subsection “Performance in LAN and WAN Settings” (see Fig. 4C).
- 7. Meta-Analysis Comparison (R1, R3, R4)**
 - a. A new subsection, “Performance Evaluation against Meta-Analysis,” presents results comparing PP-GWAS and meta-analysis across increasing numbers of nodes.
 - b. We include Manhattan plots to visualize the performance of each method as compared with the centralized reference REGENIE (see Figs. 5–7).
- 8. Enhanced Methodological Detail (R3, R4)**
 - a. The Methods section now contains a dedicated “Randomized Encoding” subsection, and also provides clearer explanations of the ADMM and CGD algorithms.
- 9. Privacy Guarantee Improvement (R1, R4)**
 - a. We utilize triple-matrix masking to eliminate reliance on secret cohort sizes, strengthening our privacy proof. The revised Privacy Analysis (Theorem 1) reflects this update.
- 10. Grammar, Style, and Minor Edits**
 - a. We fixed any typographical and grammatical errors throughout.
- 11. Code Availability (R3, R4)**
 - a. The GitHub page is now fully updated with all relevant files and information required to replicate the experiments. This includes a small simulated dataset consisting of 1000 samples, 10000 SNPs split into 2 blocks, 5 covariates, meant for analysis amongst 3 computational nodes.

All changes are reflected in the revision of the manuscript using a different font color and are detailed further in our point-by-point response.

Reviewer #2 (Remarks to the Author):

We thank the reviewer for their time and effort into reviewing our work. Although we are unsure which reviewer R2 worked with, we hope that our responses and accompanying revisions address the concerns raised and are satisfactory.

List of changes:

Here, we provide with the list of changes made in our resubmission for the reference of both the editor and the reviewers. In our list of changes, we refer to the reviewers 1,2,3 and 4 as R1, R2, R3 and R4 respectively. Since we are unsure which reviewer R2 worked with on the paper, we are unable to mention them below.

- 1. Expanded Background and Comparative Methods (R1, R3, R4)**
 - a. The Introduction now includes a more comprehensive overview of meta-analysis, S-GWAS, and SF-GWAS.
- 2. Manuscript Restructuring (R1, R4)**
 - a. The paper has been fully restructured to align with Nature Communications' formatting standards.
 - b. Content previously in the appendices, particularly in the Methods section, has been moved into the main text to improve readability.
- 3. Clarification on Quantitative Data Assumptions (R1)**
 - a. Throughout the manuscript, we now explicitly state that both PP-GWAS and SF-GWAS (built on LMM architectures) operate on quantitative phenotypes, consistent with the design of REGENIE which both methods are based on.
- 4. Updated SF-GWAS Benchmarking**
 - a. Since SF-GWAS was published very recently in Nature Genetics and had additional benchmarking, we now test the performance of PP-GWAS on datasets of varying covariate sizes, mirroring the new experiments reported in SF-GWAS (see Fig. 3C).
- 5. Large-Scale Performance Evaluation (R1)**
 - a. Additional benchmarks on UK Biobank-scale datasets have been performed utilizing more computational resources than was previously available with the help of deNBI Cloud (see Fig. 3E).
- 6. Network Condition Experiments (R3)**
 - a. We report the performance of PP-GWAS under diverse network conditions in the new subsection "Performance in LAN and WAN Settings" (see Fig. 4C).
- 7. Meta-Analysis Comparison (R1, R3, R4)**
 - a. A new subsection, "Performance Evaluation against Meta-Analysis," presents results comparing PP-GWAS and meta-analysis across increasing numbers of nodes.
 - b. We include Manhattan plots to visualize the performance of each method as compared with the centralized reference REGENIE (see Figs. 5–7).
- 8. Enhanced Methodological Detail (R3, R4)**
 - a. The Methods section now contains a dedicated "Randomized Encoding" subsection, and also provides clearer explanations of the ADMM and CGD algorithms.
- 9. Privacy Guarantee Improvement (R1, R4)**
 - a. We utilize triple-matrix masking to eliminate reliance on secret cohort sizes, strengthening our privacy proof. The revised Privacy Analysis (Theorem 1) reflects this update.

10. Grammar, Style, and Minor Edits

- a. We fixed any typographical and grammatical errors throughout.

11. Code Availability (R3, R4)

- a. The GitHub page is now fully updated with all relevant files and information required to replicate the experiments. This includes a small simulated dataset consisting of 1000 samples, 10000 SNPs split into 2 blocks, 5 covariates, meant for analysis amongst 3 computational nodes.

All changes are reflected in the revision of the manuscript using a different font color and are detailed further in our point-by-point response.

Reviewer #3 (Remarks to the Author):

We appreciate the reviewer's valuable feedback, which has greatly improved our manuscript's clarity and scientific rigor. We trust that our point-by-point responses and the accompanying revisions fully address the concerns raised.

[Technical novelty] The method section is lacking a detailed methods section, so it is difficult to evaluate the novelty of the proposed techniques in this manuscript. The authors utilize randomized encoding to obfuscate the data, which was already introduced in the previous work [19].

Thank you for this comment. We have now restructured the manuscript as a whole to reflect the expectations of Nature Communications. Instead of having all information in the appendices, we have now included them accordingly in a detailed method's section. We have also added detail on Randomized Encoding formally, and illustrated how it is used in our methods section. Although randomized encoding is a techniques utilized throughout literature (as are other techniques such as Differential Privacy, and Homomorphic Encryption which is utilized by SF-GWAS), we believe that the novelty lies in how randomized encoding is utilized in achieving an intended result - in our case - performing REGENIE in a privacy-preserving manner when data is distributed. Further, we have added an expanded description of both the ADMM and CGD based algorithms in the study for more rigor.

Additionally, critical details about the preprocessing steps are missing - specifically, how to share a secret key and use this seed for generating the subsequent shared matrices (e.g., Step 1 in Algorithm 1, Step 1 in Algorithm 3).

We thank the reviewer for raising this point. Establishing a shared pseudo-random seed is a one-time setup step that is not important to the methodological core of PP-GWAS. Any authenticated key-agreement or collective-randomness protocol that provides all collaborating nodes with the same uniformly random seed is sufficient—e.g., an (elliptic-curve) Diffie-Hellman exchange. The requirement is that the seed remain unknown to external adversaries. In our implementation we generate a single seed from the operating-system RNG, record it, and initialise every node with this value. Fixing the seed after generation ensured full reproducibility of our benchmarks and isolates the evaluation to the proposed algorithms. Practitioners can replace this one-line initialisation with any standard key-agreement protocol—without altering the remainder of the pipeline.

[Comparison] It would be beneficial to include a comparison with other cryptographic methods. For example, Homomorphic Encryption requires data providers to access a third-party with high computational resources (e.g., cloud server), and the proposed method also relies on a third party to perform the computations.

Thank you for your comment. We already compare PP-GWAS against two state-of-the-art cryptographic baselines: S-GWAS, which relies on secure multiparty computation, and SF-GWAS, which combines MPC with fully homomorphic encryption. In the revised Introduction we make this explicit. Our benchmarks thus inherently contrast PP-GWAS with these heavier cryptographic approaches. In our experiments section, we also report the RAM utility and communication costs of PP-GWAS as opposed to SF-GWAS which gives a high level overview of the computational resources, necessitated especially by the third party. We also have included a subsection discussing PP-GWAS's efficiency as compared to meta-analysis which is a long-standing technique to performing multi-site GWAS where individual level summary statistics are combined to study a combined dataset.

[Experiments] A more realistic experimental setup should be considered. Specifically, it would be helpful to incorporate multiple servers in a LAN or WAN setting environment. This would provide a more accurate estimate of the communication costs between parties.

Thank you for requesting for benchmarking on various network settings. On this end, we performed experiments on Google Cloud on both LAN and WAN settings, and have reported our observed results as opposed to SF-GWAS in our new subsection *Performance in LAN and WAN Settings* under *Experiments*.

[Source code] The source code is available, but the README file is empty, and there is no guidance on how to run the source code.

Reviewer #3 (Remarks on code availability):

The source code is available, but the README file is empty, and there is no guidance on how to run the source code.

We apologize for this oversight and thank the reviewer for pointing this out. The GitHub page is now fully updated with all relevant files and information required to replicate the experiments.

We would like to express our sincere gratitude to the reviewer for the time and effort invested in evaluating our work and providing feedback. Your suggestions and comments have been invaluable in strengthening the paper and in improving its scientific rigor. We appreciate the constructive feedback, and look forward to any further comments you may have.

List of changes:

Here, we provide with the list of changes made in our resubmission for the reference of both the editor and the reviewers. In our list of changes, we refer to the reviewers 1,2,3 and 4 as R1, R2, R3 and R4 respectively. Since we are unsure which reviewer R2 worked with on the paper, we are unable to mention them below.

- 1. Expanded Background and Comparative Methods (R1, R3, R4)**
 - a. The Introduction now includes a more comprehensive overview of meta-analysis, S-GWAS, and SF-GWAS.
- 2. Manuscript Restructuring (R1, R4)**
 - a. The paper has been fully restructured to align with Nature Communications' formatting standards.
 - b. Content previously in the appendices, particularly in the Methods section, has been moved into the main text to improve readability.
- 3. Clarification on Quantitative Data Assumptions (R1)**
 - a. Throughout the manuscript, we now explicitly state that both PP-GWAS and SF-GWAS (built on LMM architectures) operate on quantitative phenotypes, consistent with the design of REGENIE which both methods are based on.
- 4. Updated SF-GWAS Benchmarking**
 - a. Since SF-GWAS was published very recently in Nature Genetics and had additional benchmarking, we now test the performance of PP-GWAS on datasets of varying covariate sizes, mirroring the new experiments reported in SF-GWAS (see Fig. 3C).
- 5. Large-Scale Performance Evaluation (R1)**
 - a. Additional benchmarks on UK Biobank-scale datasets have been performed utilizing more computational resources than was previously available with the help of deNBI Cloud (see Fig. 3E).
- 6. Network Condition Experiments (R3)**
 - a. We report the performance of PP-GWAS under diverse network conditions in the new subsection "Performance in LAN and WAN Settings" (see Fig. 4C).
- 7. Meta-Analysis Comparison (R1, R3, R4)**

- a. A new subsection, “Performance Evaluation against Meta-Analysis,” presents results comparing PP-GWAS and meta-analysis across increasing numbers of nodes.
 - b. We include Manhattan plots to visualize the performance of each method as compared with the centralized reference REGENIE (see Figs. 5–7).
- 8. Enhanced Methodological Detail (R3, R4)**
- a. The Methods section now contains a dedicated “Randomized Encoding” subsection, and also provides clearer explanations of the ADMM and CGD algorithms.
- 9. Privacy Guarantee Improvement (R1, R4)**
- a. We utilize triple-matrix masking to eliminate reliance on secret cohort sizes, strengthening our privacy proof. The revised Privacy Analysis (Theorem 1) reflects this update.
- 10. Grammar, Style, and Minor Edits**
- a. We fixed any typographical and grammatical errors throughout.
- 11. Code Availability (R3, R4)**
- a. The GitHub page is now fully updated with all relevant files and information required to replicate the experiments. This includes a small simulated dataset consisting of 1000 samples, 10000 SNPs split into 2 blocks, 5 covariates, meant for analysis amongst 3 computational nodes.

All changes are reflected in the revision of the manuscript using a different font color and are detailed further in our point-by-point response.

Reviewer #4 (Remarks to the Author):

This paper by Swaminathan et al. introduces a method for conducting GWAS while preserving data privacy across multiple sites.

Thank you for taking your time to review the paper. We highly appreciate your comments and feedback that have improved the paper significantly, and hope that our point-by-point response below and our accompanied revisions are to your satisfaction.

Although the approach addresses an important challenge in multi-site genomic research, I found the presentation of the methodology difficult to follow. Much of the technical detail is relegated to the appendices. This makes it hard for readers to grasp the core concepts without sifting through supplementary sections.

Thank you very much for your remark. We have now restructured the manuscript as a whole to reflect the expectations of Nature Communications. Instead of having all information in the appendices, we have now included them in a detailed method's section. Further, we have both expanded on randomized encoding, and how it's used in the methodology, as well as an expanded description of both the ADMM and CGD based algorithms in the study.

One key issue with the paper is the comparison with SF-GWAS. While the authors benchmark their method against SF-GWAS, the comparison seems unfair. SF-GWAS provides two approaches—one that accounts for population stratification and another that uses a linear mixed model (LMM). The proposed PP-GWAS method is specifically designed for LMM-based analyses, so it doesn't provide a full comparison with SF-GWAS's broader capabilities. A more balanced comparison, including other secure GWAS approaches, would give a clearer perspective on the relative strengths and weaknesses of the method.

Thank you for this observation. We have now revised our Introduction to better explain S-GWAS and SF-GWAS, to further help place our work in the literature. As pointed out by the reviewer, SF-GWAS offers two workflows: one computes PCA followed by standard association testing, and the other implements a linear mixed-model (LMM) approach based on REGENIE. Our PP-GWAS pipeline focuses on the latter, performing REGENIE in a distributed, privacy-preserving manner using randomized encoding, and therefore does not currently include a PCA module. We recognize that secure PCA has potential to further strengthen LMM, and we plan to incorporate privacy-preserving PCA—leveraging randomized encoding methods as described in [1,2]—in future extensions of our framework.

Related to the above paragraph: Another critical gap in the manuscript is the absence of comparisons with other relevant secure and private GWAS methods. Important works that should be considered for benchmarking include Kockan, C., Zhu, K., Dokmai, N., et al., "Sketching algorithms for genomic data analysis and querying in a secure enclave," Nat Methods, 2020 and Li, Wentao et al., "Federated generalized linear mixed models for collaborative genome-wide association studies," iScience, 2023. I

Thank you for raising our lack of comparison to other existing studies. To the best of our knowledge, SF-GWAS was the fastest privacy-preserving GWAS algorithm that could work with large-scale data in a distributed manner. Further, we included comparison to S-GWAS as well. Paper 1 listed in your comment operates in a completely different setting and proposes a hardware-based approach to performing secure GWAS, which restricts comparison. It proposes a sketching algorithm that finds the top- k SNPs, and doesn't not perform an association test such as other standard GWAS techniques that test the association of every SNP. Furthermore, with k values just in a few hundreds, the method starts deteriorating in accuracy. Hence, we aren't able to find any meaningful metrics to compare the method with PP-GWAS, since the methods are fundamentally different. Paper 2 meanwhile suggests that the method they propose - dMega - is more suitable for small-scale validation studies or targeted association testing, where sites may focus on small regions deemed significant at each site separately and validate using new samples

collaboratively (computations on 10,000 SNPs takes 20,000s on a single thread). Hence we were unable to compare these methods. But since the concerns expressed are of our benchmarking being limited, we expanded our *Introduction* section to describe and cite other works, and performed further benchmarking against meta-analysis which is a long-standing technique to performing multi-site GWAS where individual level summary statistics are combined to study a combined dataset.

The paper claims that its core contribution is in the domain of privacy. However, the privacy analysis appears only towards the end of the appendix. It is also written in a way that makes it less accessible for a wider audience. Given that privacy is a primary focus, the authors should place a greater emphasis on demonstrating that their method ensures privacy and security. Specifically, the paper would benefit from a more detailed and accessible explanation of how the method ensures that the encodings do not inadvertently reveal genetic information about participants.

Thank you for expressing your concerns about the relegation of privacy-related discussions to the appendices. Following this, we have now restructured the paper to discuss privacy-related matters early in the paper, and further added a discussion on Randomized Encoding in the Methods section to make it accessible to a wider audience. We have also enhanced explanations in the methods section to better emphasize the usage of randomized encoding.

Furthermore, while the paper acknowledges that the security of the encoding relies on avoiding reconstruction attacks, there is a lack of discussion of this matter. A thorough analysis of the potential risks posed by reconstruction attacks is important, especially in scenarios where adversaries might have access to additional data that could be used to reverse-engineer sensitive information. The authors should provide a deeper exploration of the likelihood and impact of such attacks, and how their method mitigates these risks.

Thank you for raising this important point. We have rephrased the manuscript to more clearly articulate our privacy guarantees and underlying assumptions. To the best of our knowledge, and aside from contrived “extreme” outlier cases - no practical reconstruction attacks exist against our encoding, and we have not identified any in the literature. At the same time, we recognize that all privacy-enhancing techniques carry the risk of future attacks; we therefore explicitly acknowledge this limitation in our *Introduction*.

The use of real-world data for validation is a strength of the study; however, the paper overlooks a key limitation of its approach. The datasets used are jointly genotyped, which simplifies the QC process. In real-world scenarios, data from different sites may be independently sequenced, which presents additional challenges for QC. The paper should address how the proposed method would handle variations in sequencing quality across sites, and how this might affect the robustness of the analyses.

Thank you for this comment. We agree that quality control (QC) challenges arising from independently sequenced datasets are important considerations in large-scale genomics. However, we would like to clarify that our work focuses specifically on the analysis stage of multi-site GWAS in a privacy-preserving setting, assuming that input datasets have already undergone local QC following standard best practices, as is the case with related literature we have discussed in the paper. Issues related to sequencing quality, genotype calling, or joint vs. independent genotyping pipelines fall outside the scope of our method.

The github is not helpful. There is no information on how to install and use the code. It is just bunch of folders with code in it. I was not able evaluate the code due to this lack of explanation.

We apologize for this oversight and thank the reviewer for pointing this out. The GitHub page is now fully updated with all relevant files and information required to replicate the experiments.

Overall, while the paper tackles an important problem in the domain of privacy-preserving GWAS, it would benefit greatly from a clearer presentation of its methods, a more comprehensive privacy analysis, a fairer comparison with alternative approaches, and an exploration of real-world challenges.

Reviewer #4 (Remarks on code availability):

I am not able to run the code as there is no information on how to run it. There are also bunch of files in their results folder, which I hope does not contain any private information about their results with real-world data they used and it is only the simulated data.

We apologize for this oversight and thank the reviewer for pointing this out. The GitHub page is now fully updated with all relevant files and information required to replicate the experiments.

We would like to express our sincere gratitude to the reviewer for the time and effort invested in evaluating our work and providing feedback. Your suggestions and comments have been invaluable in strengthening the paper and in improving its scientific rigor. We appreciate the constructive feedback, and look forward to any further comments you may have.

[1] Hannemann, A., Ünal, A.B., Swaminathan, A., Buchmann, E., Akgün, M.: A privacy-preserving framework for collaborative machine learning with kernel methods

[2] Hannemann, A., Swaminathan, A., Ünal, A.B., Akgün, M.: Private, efficient and scalable kernel learning for medical image analysis

List of changes:

Here, we provide with the list of changes made in our resubmission for the reference of both the editor and the reviewers. In our list of changes, we refer to the reviewers 1,2,3 and 4 as R1, R2, R3 and R4 respectively. Since we are unsure which reviewer R2 worked with on the paper, we are unable to mention them below.

- 1. Expanded Background and Comparative Methods (R1, R3, R4)**
 - a. The Introduction now includes a more comprehensive overview of meta-analysis, S-GWAS, and SF-GWAS.
- 2. Manuscript Restructuring (R1, R4)**
 - a. The paper has been fully restructured to align with Nature Communications' formatting standards.
 - b. Content previously in the appendices, particularly in the Methods section, has been moved into the main text to improve readability.
- 3. Clarification on Quantitative Data Assumptions (R1)**
 - a. Throughout the manuscript, we now explicitly state that both PP-GWAS and SF-GWAS (built on LMM architectures) operate on quantitative phenotypes, consistent with the design of REGENIE which both methods are based on.
- 4. Updated SF-GWAS Benchmarking**
 - a. Since SF-GWAS was published very recently in Nature Genetics and had additional benchmarking, we now test the performance of PP-GWAS on datasets of varying covariate sizes, mirroring the new experiments reported in SF-GWAS (see Fig. 3C).
- 5. Large-Scale Performance Evaluation (R1)**
 - a. Additional benchmarks on UK Biobank-scale datasets have been performed utilizing more computational resources than was previously available with the help of deNBI Cloud (see Fig. 3E).

6. Network Condition Experiments (R3)

- a. We report the performance of PP-GWAS under diverse network conditions in the new subsection “Performance in LAN and WAN Settings” (see Fig. 4C).

7. Meta-Analysis Comparison (R1, R3, R4)

- a. A new subsection, “Performance Evaluation against Meta-Analysis,” presents results comparing PP-GWAS and meta-analysis across increasing numbers of nodes.
- b. We include Manhattan plots to visualize the performance of each method as compared with the centralized reference REGENIE (see Figs. 5–7).

8. Enhanced Methodological Detail (R3, R4)

- a. The Methods section now contains a dedicated “Randomized Encoding” subsection, and also provides clearer explanations of the ADMM and CGD algorithms.

9. Privacy Guarantee Improvement (R1, R4)

- a. We utilize triple-matrix masking to eliminate reliance on secret cohort sizes, strengthening our privacy proof. The revised Privacy Analysis (Theorem 1) reflects this update.

10. Grammar, Style, and Minor Edits

- a. We fixed any typographical and grammatical errors throughout.

11. Code Availability (R3, R4)

- a. The GitHub page is now fully updated with all relevant files and information required to replicate the experiments. This includes a small simulated dataset consisting of 1000 samples, 10000 SNPs split into 2 blocks, 5 covariates, meant for analysis amongst 3 computational nodes.

All changes are reflected in the revision of the manuscript using a different font color and are detailed further in our point-by-point response.

Reviewer #1 (Remarks to the Author):

The authors have addressed all my comments.

We thank the reviewer for taking their time to go through our revisions, and we are delighted that the revisions are satisfactory, and that we were able to address all of your concerns and comments. Thanks again for all the questions and remarks that prompted changes that improved the manuscript.

List of changes:

Here, we provide with the list of changes made in our resubmission for the reference of both the editor and the reviewers. In our list of changes, we refer to the reviewers 1,2,3 and 4 as R1, R2, R3 and R4 respectively.

1. **Supplementary Material on Accuracy Results (R3)**
 - a. We have included an appendix where we report all the accuracy scores from our tests on simulated data.
2. **Clarifications across the Paper (R3, R4)**
 - a. We have included more detail in the paper in certain sections to avoid misunderstandings, and to address the concerns raised by R3 and R4 regarding aggregated noise removal, p-value threshold computation, joint-genotyping and quality control.
3. **Toy Code (R4)**
 - a. We have updated the code repository to now include a folder “local_run” where a Jupyter notebook is available to run a small test run of the algorithm on any device (no MKL dependency needed).
4. **Mislabeling Error**
 - a. In Fig 2, we had mislabeled the datasets, and have now fixed this oversight.

All changes are reflected in the revision of the manuscript using a different font color (red for changes from initial revision, and violet for the most recent revisions) and are detailed further in our point-by-point response.

Reviewer #2 (Remarks to the Author):

We thank the reviewer for their time and effort into reviewing our work. Although we are unsure which reviewer R2 worked with, we hope that our responses and accompanying revisions address the concerns raised and are satisfactory.

List of changes:

Here, we provide with the list of changes made in our resubmission for the reference of both the editor and the reviewers. In our list of changes, we refer to the reviewers 1,2,3 and 4 as R1, R2, R3 and R4 respectively.

- 1. Supplementary Material on Accuracy Results (R3)**
 - a. We have included an appendix where we report all the accuracy scores from our tests on simulated data.
- 2. Clarifications across the Paper (R3, R4)**
 - a. We have included more detail in the paper in certain sections to avoid misunderstandings, and to address the concerns raised by R3 and R4 regarding aggregated noise removal, p-value threshold computation, joint-genotyping and quality control.
- 3. Toy Code (R4)**
 - a. We have updated the code repository to now include a folder “local_run” where a Jupyter notebook is available to run a small test run of the algorithm on any device (no MKL dependency needed).
- 4. Mislabeling Error**
 - a. In Fig 2, we had mislabeled the datasets, and have now fixed this oversight.

All changes are reflected in the revision of the manuscript using a different font color (red for changes from initial revision, and violet for the most recent revisions) and are detailed further in our point-by-point response.

Reviewer #3 (Remarks to the Author):

I think the authors have addressed my concerns in this revision.

We want to thank the reviewer for taking their time in reviewing our paper and our revisions. We are delighted to learn that we were able to address your concerns.

I could not find detailed accuracy results on the synthetic dataset.

We apologize for this omission. The synthetic-data accuracy ($r^2 = 1.00$) matches the values reported for all other datasets, and we felt that it was redundant to report the same, especially for synthetic datasets. We have now added a collection of these results to the Appendices for completeness.

Additional clarification is needed on the following points:

• How do the participating nodes remove the aggregated noise (i.e., the summation of r_i in line 483)?

Each client holds its own random mask r_i , and they collectively know the $\text{sum}(r_i)$. After the server returns the masked aggregate, the clients subtract $\text{sum}(r_i)$ to recover the true result. We have clarified this in the Methods section.

• How is the comparison with a specific threshold performed to determine whether a trait is above or below a given metric? Is this step executed locally using the total counts?

In our framework, the server's final output is the set of SNP-wise p-values. The computational nodes then apply thresholds using standard criteria (e.g., Bonferroni correction) on these p-values locally. If hiding the p-values from the server is required, the nodes can randomize the SNP order before thresholding without affecting correctness. We have expanded the "Distributed Single SNP Association Testing" subsection appropriately.

Furthermore, in the comparison with SF-GWAS, is the PCA step used in SF-GWAS excluded from the evaluation?

The SF-GWAS methodology proposed two methods, one for PCA and one for LMMs. The LMM-based approach is under-reported in their paper, and is reported to be much slower than the PCA-based approach. SF-GWAS only reports the PCA-based timings. Further, persistent issues in the publicly available SF-GWAS code required repeated debugging, and despite multiple emails, we received no response or support from its authors. We have not been able to replicate SF-GWAS on our computational resources and have only been able to compare with the results claimed in their paper. Our method is an LMM based approach to performing GWAS (similar to SF-GWAS' LMM based approach which also follows REGENIE like us). We are considering working on PCA-based GWAS later in future work since there is promise of faster computations using randomized encoding.

It would be helpful to include the detailed sizes of the datasets when they are first introduced—for example, in the subsections "Synthetic Data Generation" and "Real Datasets."

Thank you for the comment. We have now included the dataset sizes whenever we introduce the datasets.

We again appreciate the reviewer's valuable feedback, which has greatly improved our manuscript's clarity and scientific rigor. We trust that our point-by-point responses and the accompanying revisions fully address the concerns raised.

List of changes:

Here, we provide with the list of changes made in our resubmission for the reference of both the editor and the reviewers. In our list of changes, we refer to the reviewers 1,2,3 and 4 as R1, R2, R3 and R4 respectively.

- 1. Supplementary Material on Accuracy Results (R3)**
 - a. We have included an appendix where we report all the accuracy scores from our tests on simulated data.
- 2. Clarifications across the Paper (R3, R4)**
 - a. We have included more detail in the paper in certain sections to avoid misunderstandings, and to address the concerns raised by R3 and R4 regarding aggregated noise removal, p-value threshold computation, joint-genotyping and quality control.
- 3. Toy Code (R4)**
 - a. We have updated the code repository to now include a folder “local_run” where a Jupyter notebook is available to run a small test run of the algorithm on any device (no MKL dependency needed).
- 4. Mislabeling Error**
 - a. In Fig 2, we had mislabeled the datasets, and have now fixed this oversight.

All changes are reflected in the revision of the manuscript using a different font color (red for changes from initial revision, and violet for the most recent revisions) and are detailed further in our point-by-point response.

Reviewer #4 (Remarks to the Author):

I believe the authors may have misunderstood my earlier point regarding joint genotyping. When I referred to the SNP data being "jointly genotyped," I was not commenting on quality control procedures. I fully acknowledge that quality control will be conducted independently by each institution before the GWAS step. However, joint genotyping is a distinct process and should not be conflated with quality control. Joint genotyping involves calling variants across multiple samples simultaneously, which improves consistency and reduces batch effects by leveraging population-level information to better distinguish true variants from sequencing or alignment artifacts. In a federated or distributed setting, joint genotyping is not feasible because raw sequencing data cannot be shared across institutions due to privacy and governance constraints. Consequently, the authors must work with SNP data that have been independently genotyped at each site. This has important implications for their framework. Independent genotyping can introduce batch effects and inconsistencies in variant calls across sites, potentially affecting downstream analyses. Therefore, the authors should either (1) demonstrate that their method includes a mechanism to correct for this type of batch effect, or (2) provide evidence that the absence of joint genotyping does not substantially impact the accuracy or robustness of their framework.

Thank you very much for clarifying your previous remark regarding joint genotyping. We mistakenly responded regarding quality control. We agree that true joint genotyping cannot occur when raw sequencing data remain at each site. Both in our work, and in related literature (FAMHE [1], S-GWAS [2], SF-GWAS [3] - now published in Nature Genetics), the methods begin with jointly genotyped datasets that are partitioned across sites. For comparative reasons, we follow the same procedure as related literature. To reflect realistic federated scenarios, we have added the following clarification to the "Quality Control" subsection:

"In the absence of joint genotyping, data harmonization is required to identify a common set of SNPs across sites. This can be achieved by sharing only non-private information such as genomic positions, reference, alternate alleles, strand information and when available, the rsID, so an aggregator can build a common SNP list without exposing individual level data."

Additionally, our framework performs Quality Control via global statistics computed across sites, ensuring rare variants retained at one site are not inadvertently filtered out elsewhere. We have emphasized this in the same "Quality Control" subsection. Further, our approach already incorporates covariate projection to control for population structure and site-specific effects. We have added further clarification to the "Distributed Projection of Covariates and Standardizing" subsection.

The code is not intuitive to review, as it requires setting up ports and configuring the framework. For reproducibility and easier code review, it would have been preferable to provide a single, streamlined workflow along with example toy data that can be run locally.

Thank you for your comment regarding the code. We now included an additional folder in our code titled "local_run" that includes a notebook with a straightforward pipeline that sets up sockets to a default port of 8000, and launches separate python files for the server and the computational nodes. The notebook also plots the associated Manhattan plot. Note that since this was rewritten and streamlined for local testing, we removed the MKL library dependency so it can run on any device. However the time taken to do the GWAS would be slightly higher than when using MKL.

We again appreciate the reviewer's valuable feedback, which has greatly improved our manuscript's clarity and scientific rigor. We trust and hope that our point-by-point responses and the accompanying revisions fully address the concerns raised.

List of changes:

Here, we provide with the list of changes made in our resubmission for the reference of both the editor and the reviewers. In our list of changes, we refer to the reviewers 1,2,3 and 4 as R1, R2, R3 and R4 respectively.

1. **Supplementary Material on Accuracy Results (R3)**
 - a. We have included an appendix where we report all the accuracy scores from our tests on simulated data.
2. **Clarifications across the Paper (R3, R4)**
 - a. We have included more detail in the paper in certain sections to avoid misunderstandings, and to address the concerns raised by R3 and R4 regarding aggregated noise removal, p-value threshold computation, joint-genotyping and quality control.
3. **Toy Code (R4)**
 - a. We have updated the code repository to now include a folder “local_run” where a Jupyter notebook is available to run a small test run of the algorithm on any device (no MKL dependency needed).
4. **Mislabeling Error**
 - a. In Fig 2, we had mislabeled the datasets, and have now fixed this oversight.

All changes are reflected in the revision of the manuscript using a different font color (red for changes from initial revision, and violet for the most recent revisions) and are detailed further in our point-by-point response.

Response to Reviewer #4

We thank you for your insightful comments, which have helped improve the clarity of our manuscript and the usability of our code. Thank you very much for highlighting the importance of joint genotyping and initiating a discussion regarding the same.

Reviewer #4 (Remarks to the Author):

I don't believe it's sufficient to address the joint genotyping issue by simply noting that prior work also did not consider it. Since I wasn't a reviewer on the earlier paper, I can't speak to the specifics of its review process or criteria.

We agree that our choice of omitting experimentation on joint genotyping warrants a more detailed explanation and justification, and its omission in prior literature isn't a fully justified response to your concerns.

More importantly, if SNPs are jointly genotyped, it raises the question of why a federated GWAS approach is needed—wouldn't that imply the genotypes are already centrally available? On the other hand, if SNPs are not jointly genotyped, could this introduce batch effects or other artifacts? Is there a way to quantify or correct for this? It would be helpful if the authors could include a simple experiment to illustrate whether and how this impacts the results. Do the findings change meaningfully when joint genotyping is or isn't performed?

We agree joint genotyping can be critical under specific conditions [1] such as

- *low coverage*
- *discovery of very rare sites* (since cohort-wide likelihoods can rescue genotypes at sites with sparse evidence).

Regarding coverage-related issues: Modern GWAS typically rely on deep-coverage WGS/WES or arrays, followed by rigorous per-cohort quality control and imputation to large reference panels. In this setting, per-study calling plus meta-analysis performs on par with joint calling for single-variant tests. In a head-to-head study using deep-coverage data, $\geq 97\%$ of rare SNVs were detected by both strategies, non-reference genotype concordance was $\sim 99 - 99.7\%$, and association results from meta-analysis closely matched joint analysis; material discrepancies appeared primarily at $\sim 5\times$ low coverage [2]. Moreover, state of the art variant callers and pipelines (e.g., GATK HaplotypeCaller [3], DeepVariant [4], DRAGEN [5]) achieve very high accuracy on benchmark genomes (SNP/indel F1 often ≥ 0.99 at deep coverage), further reducing any marginal advantage of joint genotyping in this context [6,7,8]. Standard GWAS practice also emphasizes per-cohort QC, imputation (HRC/TOPMed and other large panels), and meta-analysis, which harmonize variants across studies and mitigate batch effects without centralizing raw genotypes [9,10,11].

Regarding rescuing sparse SNPs: Our cross-cohort quality control step and harmonization ensure that variants observed only in a subset of cohorts are not discarded simply because they are absent at a single computational node. Practically, this delivers the same downstream advantage of joint genotyping, without requiring data centralization.

Accordingly, based on established practice and empirical evidence in deep-coverage settings, and our method's ability to rescue rare SNPs absent in some nodes, we believe it is justified to omit extensive experimentation on joint genotyping in our current GWAS pipeline since joint genotyping doesn't provide any tangible benefits to our work.

However, we recognize the need to expand on this omission explicitly in the manuscript and hence have updated the manuscript with the following text in the Quality Control subsection under Methods.

PP-GWAS does not necessitate joint genotyping, which requires centralisation of data. This is due to two reasons: first, modern variant-calling and imputation pipelines typically operate at high coverage and achieve high accuracy, making the marginal value of joint genotyping small for single-variant association. Second, our globally performed QC retains variants present in any participating site, so rare variants that might otherwise be discarded by site-specific QC are preserved. This realizes a principal benefit of joint genotyping where rare SNPs absent at a cohort will be "rescued". When using our method in collaborative settings, in the absence of joint genotyping, data harmonization is required to identify a common set of SNPs across sites. This can be achieved by sharing only non-private information such as genomic positions, reference, alternate alleles, strand information and when available, the rsID, so an aggregator can build a common SNP list without exposing individual level data.

I'm having trouble running the local_run. To start with, there's no README or documentation explaining what the components are or how to use them. I managed to make some partial progress using Google and ChatGPT, but the process was far from straightforward. There were no instructions on how to properly set up the conda environment required for the local run. Ultimately, I got stuck on an issue where the script failed to generate the neg_log_transfer.npy file, and I wasn't able to resolve it.

We apologize for the inconvenience caused by our oversight. The GitHub repository has now been updated with a detailed README, environment files, and step-by-step usage documentation for a local test run with small sample counts. We have also now verified that the code successfully generates the neg_log_transfer.npy file.

We thank you again for your time and effort in reviewing our work, and for raising your concerns throughout the review process. We hope that our response is satisfactory and that we have not misunderstood any of the concerns or comments made.

[1]

<https://gatk.broadinstitute.org/hc/en-us/articles/360035890431-The-logic-of-joint-calling-for-germline-short-variants>

[2] Chen, Zhongsheng, Michael Boehnke, and Christian Fuchsberger. "Combining sequence data from multiple studies: Impact of analysis strategies on rare variant calling and association results." *Genetic epidemiology* 44.1 (2020): 41-51.

[3] Poplin, Ryan, et al. "Scaling accurate genetic variant discovery to tens of thousands of samples." *bioRxiv* (2017): 201178.

[4] Poplin, Ryan, et al. "A universal SNP and small-indel variant caller using deep neural networks." *Nature biotechnology* 36.10 (2018): 983-987.

[5] Behera, Sairam, et al. "Comprehensive genome analysis and variant detection at scale using DRAGEN." *Nature Biotechnology* 43.7 (2025): 1177-1191.

[6] Koboldt, Daniel C. "Best practices for variant calling in clinical sequencing." *Genome medicine* 12.1 (2020): 91.

[7] Zhao, Sen, et al. "Accuracy and efficiency of germline variant calling pipelines for human genome data." *Scientific reports* 10.1 (2020): 20222.

[8] Kosugi, Shunichi, and Chikashi Terao. "Comparative evaluation of SNVs, indels, and structural variations detected with short-and long-read sequencing data." *Human genome variation* 11.1 (2024): 18.

[9] Uffelmann, Emil, et al. "Genome-wide association studies." *Nature Reviews Methods Primers* 1.1 (2021): 59.

[10] Sung, Yun Ju, et al. "An empirical comparison of meta-analysis and mega-analysis of individual participant data for identifying gene-environment interactions." *Genetic epidemiology* 38.4 (2014): 369-378.

[11] Kowalski, Madeline H., et al. "Use of > 100,000 NHLBI Trans-Omics for Precision Medicine (TOPMed) Consortium whole genome sequences improves imputation quality and detection of rare variant associations in admixed African and Hispanic/Latino populations." *PLoS genetics* 15.12 (2019): e1008500.

List of changes:

Here, we provide with the list of changes made in our resubmission for the reference of both the editor and the reviewers. In our list of changes, we refer to the reviewer 4 as R4.

1. **Toy Code update (R4)**
 - a. The local run folder on GitHub now has a README file with explicit instructions on how to run the code. Further, an environment file has been provided to successfully run the code.
2. **Joint Genotyping - Manuscript update (R4)**
 - a. The Quality Control subsection under the Methods section has been updated to explicitly address why joint-genotyping does not provide any tangible benefits to our work.

All changes are reflected in the revision of the manuscript using a different font color (red for changes from first revision, violet for the second and olive for the most recent revisions).

Dear Reviewer #4

We thank you for your time, effort and comments, which allow us to clarify a critical distinction and to improve the scientific rigor of our work.

Thank you for your detailed response. However, I remain unconvinced that it addresses my key concern. Even the cited literature clearly shows that both the genotyping pipeline and sequencing platform strongly influence SNP distributions (see below for deeper dive).

In this context, it's also important to clarify that joint genotyping is not merely the process of finding common variants across samples and imputing those absent in one dataset from another.

Rather, its strength lies in performing variant calling within a unified framework—using the same pipeline across samples while explicitly considering sequencing platform as a covariate. This approach allows systematic biases introduced by different technologies to be properly accounted for, ensuring that variant calls are biologically meaningful rather than artifacts of machine or pipeline choice.

Therefore, you must either:

Empirically demonstrate that platform or pipeline effects are neutralized in your tool, or Include them as covariates, with a clear mechanism for correcting potential biases.

You're right that including platform/pipeline information as covariate information is a recommended approach to correct for technical artefacts (even in centralized studies where data is pooled from different sites). And this is precisely what we do in PP-GWAS already (the second option of what you proposed). As per your suggestion to include technical information as covariates, we want to highlight that we already do this when we perform covariate projection at the beginning of the algorithm. However, we thank you for bringing it to our notice that we did not make this clear in our manuscript. In fact, the real-world bladder cancer dataset we tested PP-GWAS on included technical/study metadata that we used to adjust for technical artefacts (genotyping platforms varied by study center). We therefore included site indicator variables, which absorb platform differences. We have now made this explicit in the manuscript (refer to page 6).

On the other hand, addressing technical heterogeneity at the variant-calling stage is a separate challenge within the domain of privacy-preserving variant calling. Our contribution, however, is a privacy-preserving GWAS framework. For the analysis phase, the established and sufficient method to correct for residual batch effects is to include them as covariates, rightly as you suggested. This is precisely the standard that REGENIE, and by extension PP-GWAS, are designed to meet.

The exact mechanism to include technical information as covariate data is a case-by-case setting depending on how the data was sampled, but one can follow standard practice such as using one-hot encodings. We agree that this needs to be addressed in the manuscript for better clarity, and hence have added the following text to Methods to make this more explicit (refer to page 19).

In settings where cohorts differ by sequencing platform, or variant-calling pipeline, each node can encode platform, pipeline, batch indicators as covariates to correct for potential artefacts as is the standard across various studies [1-7].

Below is relevant literature for context:

Chen et al. conducted a systematic comparison of 27 sequencer–variant caller combinations (including Illumina’s NovaSeq, BGISEQ-500, MGISEQ-2000, using pipelines such as GATK-HC, Strelka2, and Samtools-VarScan2). This work found that although SNP calling F-scores remained high (>0.96 for WES, >0.975 for WGS), INDEL calling was far more variable (F-scores 0.71–0.93), highlighting diverging variant profiles across platform–pipeline combinations. Citation: Chen, J., Li, X., Zhong, H. et al. Systematic comparison of germline variant calling pipelines cross multiple next-generation sequencers. *Sci Rep* 9, 9345 (2019). <https://doi.org/10.1038/s41598-019-45835-3>

Foxx et al. compared germline variant callers across platforms including BGISEQ-500, MGISEQ-2000, and NovaSeq. They reported that DeepVariant consistently exhibited the highest accuracy, while GATK and Sentieon lagged behind—underlining the influence of both machine and software on variant outcomes. Citation: Foxx J, Tighe SW, Nicolet CM, Zook JM, Byrska-Bishop M, Clarke WE, Khayat MM, Mahmoud M, Laaguiby PK, Herbert ZT, Warner D, Grills GS, Jen J, Levy S, Xiang J, Alonso A, Zhao X, Zhang W, Teng F, Zhao Y, Lu H, Schroth GP, Narzisi G, Farmerie W, Sedlazeck FJ, Baldwin DA, Mason CE. Performance assessment of DNA sequencing platforms in the ABRF Next-Generation Sequencing Study. *Nat Biotechnol.* 2021 Sep;39(9):1129-1140. doi: 10.1038/s41587-021-01049-5. Epub 2021 Sep 9. Erratum in: *Nat Biotechnol.* 2021 Nov;39(11):1466. doi: 10.1038/s41587-021-01122-z. PMID: 34504351; PMCID: PMC8985210.

Abdelwahab et al. benchmarked conventional versus AI-based variant callers (e.g., BCFTools, GATK4, Platypus, DNAscope, DeepVariant) across Illumina, PacBio HiFi, and Oxford Nanopore data. They found that AI-based tools generally outperformed traditional ones, evidencing how platform error profiles and calling algorithms dictate variant call reliability. Citation: Abdelwahab, O., Belzile, F. & Torkamaneh, D. Performance analysis of conventional and AI-based variant callers using short and long reads. *BMC Bioinformatics* 24, 472 (2023). <https://doi.org/10.1186/s12859-023-05596-3>

[1] Mbatchou, Joelle, et al. "Computationally efficient whole-genome regression for quantitative and binary traits." *Nature genetics* 53.7 (2021): 1097-1103.

[2] Loh, Po-Ru, et al. "Mixed-model association for biobank-scale datasets." *Nature genetics* 50.7 (2018): 906-908.

[3] Price, Alkes L., et al. "Principal components analysis corrects for stratification in genome-wide association studies." *Nature genetics* 38.8 (2006): 904-909.

[4] Winkler, Thomas W., et al. "Quality control and conduct of genome-wide association meta-analyses." *Nature protocols* 9.5 (2014): 1192-1212.

[5] Canela-Xandri, Oriol, Konrad Rawlik, and Albert Tenesa. "An atlas of genetic associations in UK Biobank." *Nature genetics* 50.11 (2018): 1593-1599.

[6] Horikoshi, Momoko, et al. "Genome-wide associations for birth weight and correlations with adult disease." *Nature* 538.7624 (2016): 248-252.

[7] Cole, Joanne B., Jose C. Florez, and Joel N. Hirschhorn. "Comprehensive genomic analysis of dietary habits in UK Biobank identifies hundreds of genetic associations." *Nature communications* 11.1 (2020): 1467.

Additionally, I still cannot run the tool—please refer to the Code section for details on the errors I am encountering. The code is still not running for me despite trying multiple approaches. To start, the environment file for the local run specifies the environment name as "ppgwas_test", but the README instructs users to activate "ppgwas-env". I adjusted it to "ppgwas_test", but the environment still lacked several essential dependencies—including jupyter, numpy, matplotlib, and pysnpools. This suggests there's an underlying issue with the environment setup that I wasn't able to debug.

I attempted to work around this by manually installing the missing dependencies, but even then, I encountered persistent errors. I tried running the code in several ways (directly with Python and within a Jupyter notebook), and in both cases I received errors. I've copied below the error output I received from the notebook run.

Below it seems like the .sh file ran:

```
"Launching server on port 8000 (logs → logs/ppgwas_20250823_090756/server)
```

```
Spawning 4 clients (logs → logs/ppgwas_20250823_090756/clients)
```

```
All processes finished. Logs in logs/ppgwas_20250823_090756"
```

However, the error in the next block suggests that it is still lacking the log file

```
"-----"
```

```
FileNotFoundError Traceback (most recent call last)
```

```
Cell In[8], line 5
```

```
2 import matplotlib.pyplot as plt
```

```
4 file_path = f"Data/N{N}_M{M}_C{C}_P{P}_B{B}/neg_log_transfer.npy"
```

```
-- 5 neg_log_p = np.load(file_path)
```

```
6 positions = np.arange(len(neg_log_p))
```

```
8 plt.figure(figsize=(10, 5))
```

```
File ~/miniconda3/envs/ppgwas_test/lib/python3.8/site-packages/numpy/lib/npio.py:405, in load(file, mmap_mode, allow_pickle, fix_imports, encoding, max_header_size)
```

```
403 own_fid = False
404 else:
 405 fid = stack.enter_context(open(os_fspath(file), "rb"))
406 own_fid = True
408 # Code to distinguish from NumPy binary files and pickles.
FileNotFoundError: [Errno 2] No such file or directory:
'Data/N10000_M15000_C10_P4_B2/neg_log_transfer.npy'
```

We sincerely apologize for the inconvenience and frustration encountered during the installation process. From your write up, our best guess now is that despite creating the environment, it was not used in the notebook where you ran the code. This would explain why despite the environment file containing the packages you mentioned, they were seemingly not installed. Please ensure that the environment is used since the interdependency between different versions of the packages may cause issues when you manually install them. This can be done by selecting the environment as the Jupyter kernel (or as the IDE interpreter) before running the notebook. We have now made this explicit in the README file as well.

We are committed to ensuring the code is easily reproducible. We tested the provided code ourselves on multiple independent machines. If you continue to face issues, could you please provide the exact error logs from all the folders? This will allow us to identify if there is a missing dependency or a specific system configuration we need to account for in our documentation. Further, for reproducibility and transparency, we have now provided the files we generated from a test run we ran along with our resubmission. This includes all the input data, log files and the notebook where we ran the code.

We would like to extend our gratitude to you once again for your thorough review, and your questions. We hope that our point-by-point response addresses all your remaining concerns.

List of changes:

Here, we provide with the list of changes made in our resubmission for the reference of both the editor and the reviewers. In our list of changes, we refer to the reviewer 4 as R4.

1. Covariate Projection (R4)

- a. The Quality Control subsection under the Methods section has been updated to address how covariate correction could help mitigate technical artefacts when nodes use different technologies.
- b. We added a clarification of available covariates for the real-world datasets.

All changes are reflected in the revision of the manuscript using a different font color (red for changes from first revision, violet for the second, olive for the third and blue for the most recent revisions).

Dear Reviewer #4

We thank you for your time, effort and comments, which allow us to improve the scientific rigor and reproducibility of our work. We are happy to submit this final revision with all the suggestions incorporated, and the code fixed.

I appreciate the author's detailed response to my concerns. However, I do not see these limitations reflected in the discussion section of the revised manuscript. As it stands, the discussion reads more like a promotion of the study and its advantages over other methods, rather than a balanced assessment. I strongly urge the authors to explicitly state the limitations that were acknowledged in their response and raised in my initial concerns. In particular, it should be made clear that:

- * There may be issues when genotypes are derived from non-joint genotyping across different sites.
- * The method would be most effective if implemented in the context of a privacy-preserving, federated joint genotyping framework.
- * It is unrealistic to expect clients to share genotypes, perform joint genotyping, and then return to a privacy-preserving GWAS model. Once genotypes are shared, the core rationale for federated GWAS is undermined.

Thank you for this important observation. Although we addressed the limitations in the main text, we agree it is essential to include them in the Discussion. We have now added a dedicated subsection, "Limitations," within the Discussion that covers these points. You can find the following text in the Limitations subsection:

PP-GWAS operates on datasets that may be generated by different sites without joint-genotyping. In such settings, platform- and pipeline-specific biases can induce variant-level discrepancies. We mitigate global batch effects via harmonization (shared positions, alleles, strand, and rsIDs), perform global quality control that retains rare variants present at any participating site, and remove covariate effects using covariate projection which includes site, platform/pipeline, and batch indicators. These steps, which are standard even in centralized analyses where data is pooled from different sources, are effective for single-variant association but do not eliminate all effects of technical heterogeneity.

PP-GWAS, as well as other state-of-the-art privacy-preserving distributed GWAS would be most effective when upstream variant calls are produced within a unified framework. The privacy-preserving way to achieve this is a distributed joint-genotyping layer that accounts for platform differences during variant calling without centralising raw data. Designing such a layer e.g., using secure aggregation, multi-party computation, homomorphic encryption, or trusted hardware remains an important direction for future research.

Finally, we do not advocate centralising or sharing raw genotypes for joint genotyping and then returning to a privacy-preserving distributed GWAS workflow. Were genotypes to be shared, the core rationale for privacy-preserving analyses would be undermined. PP-GWAS is therefore intended either (i) for non-jointly genotyped settings with the above mitigations and explicit technical covariates, acknowledging residual confounding may persist, or (ii) to be composed with a privacy-preserving distributed joint-genotyping layer.

Further, we have added similar clarity to the Quality Control subsection under the Methods section.

Regarding the code, I must clarify that the problem was not with the kernel. I verified I was using the correct kernel. Upon detailed debugging, I found that the ppgwas.sh script attempts to activate a Conda environment (conda activate ppREGENIE) that does not exist in the local_run directory. Consequently, the environment fails to recognize the python command. Moreover, there is no environment file provided to create this ppREGENIE environment in the local_run folder. I attempted to substitute with ppgwas_test, but the job has been running for a very long time without clear indication of success, so it remains unclear whether this workaround is

effective. (I also tried it by reducing number samples, SNPs and covariates to see if it was a runtime issue but it was still hanging and not finishing)

In my view, the repository requires a thorough cleanup and review by the authors themselves. A well-documented README, along with consistent environment files and validation of the code in a fresh setup (after removing all of their already installed environments), is essential to ensure reproducibility and usability for others.

We sincerely apologize for the inconvenience and frustration you experienced running the ‘local_run’ implementation. After a thorough cleanup and review across multiple devices and by multiple contributors, we have verified that the code now works as intended. We identified that while the code functioned reliably on macOS and GitHub Codespaces, and the cluster workflow ran successfully on several clusters (University Cluster, deNBI Cluster, Google Cloud Platform), it did not run cleanly on Linux and Windows (an environment variable `QT_XCB_GL` was shutting the server down without any warnings / there were package dependency related errors).

We have implemented the necessary fixes and re-validated on diverse machines and setups with assistance from several colleagues (now acknowledged in the acknowledgements section). We have also substantially reviewed and rewritten the GitHub repository to address the sources of confusion and errors.

To further improve reproducibility, we added a one-click GUI that performs both data generation and PP-GWAS in a single workflow based on user inputs. We have also included a video tutorial covering both the notebook and GUI workflows in the local_run folder (along with timestamps for setup, notebook and GUI sections).

Please see the updated repository for details.

Thank you again for your time and effort in reviewing our work.

List of changes:

Here, we provide with the list of changes made in our resubmission for the reference of both the editor and the reviewers. In our list of changes, we refer to the reviewer 4 as R4.

1. Discussion Section (R4)

- a. The discussion section now includes a Limitations subsection including all discussions from prior rebuttals and reviewer (R4) comments.
- b. The Quality Control subsection has been reworded to reflect the same.

2. Repository Cleanup (R4)

- a. The GitHub repository has been thoroughly cleaned up, with necessary code changes to enable the code to run smoothly across different devices. We also have included an option to run the local_run version of the code through a GUI.
- b. A concise video tutorial on how to run the code is now in the local_run folder.

All changes are reflected in the revision of the manuscript using a different font color (red for changes from the first revision, violet for the second, olive for the third, blue for the fourth and brown for the most recent revisions).